# Denoising Diffusions with Optimal Transport: Localization, Curvature, and Multi-Scale Complexity

**Tengyuan Liang**                                    *tengyuan.liang@chicagobooth.edu*
*University of Chicago, Booth School of Business*

**Kulunu Dharmakeerthi**                                    *kulunud@uchicago.edu*
*University of Chicago*

**Takuya Koriyama**                                    *tkoriyam@chicagobooth.edu*
*University of Chicago, Booth School of Business*

**Reviewed on OpenReview:** *https://openreview.net/forum?id=sj1wU6gBXH*

## Abstract

Adding noise is easy; what about denoising? Diffusion is easy; what about reverting a diffusion? Diffusion-based generative models aim to denoise a Langevin diffusion chain, moving from a log-concave equilibrium measure $\nu$, say an isotropic Gaussian, back to a complex, possibly non-log-concave initial measure $\mu$. The score function performs denoising, moving backward in time, and predicting the conditional mean of the past location given the current one. We show that score denoising is the optimal backward map in transportation cost. What is its localization uncertainty? We show that the curvature function determines this localization uncertainty, measured as the conditional variance of the past location given the current. We study in this paper the effectiveness of the diffuse-then-denoise process: the contraction of the forward diffusion chain, offset by the possible expansion of the backward denoising chain, governs the denoising difficulty. For any initial measure $\mu$, we prove that this offset net contraction at time $t$ is characterized by the curvature complexity of a smoothed $\mu$ at a specific signal-to-noise ratio (SNR) scale $\mathsf{r}(t)$. We discover that the multi-scale curvature complexity collectively determines the difficulty of the denoising chain. Our multi-scale complexity quantifies a fine-grained notion of average-case curvature instead of the worst-case. Curiously, it depends on an integrated tail function, measuring the relative mass of locations with positive curvature versus those with negative curvature; denoising at a specific SNR scale is easy if such an integrated tail is light. We conclude with several non-log-concave examples to demonstrate how the multi-scale complexity probes the bottleneck SNR for the diffuse-then-denoise process.

## 1 Introduction

Empirically, diffusion models exhibit compelling performance as probabilistic generative models for complex, multi-dimensional probability measures (Ho et al., 2020; Sohl-Dickstein et al., 2015; Song & Ermon, 2019; Song et al., 2021; Karras et al., 2022). They are often employed when traditional sampling methods suffer, such as when the probability measure is multi-modal and supported on an unknown manifold that is hard to mathematize, such as the probability distribution of (pixels of) images. Despite their impressive performance in practice, several fundamental theoretical questions regarding denoising quality remain unanswered (Block et al., 2020; Chen et al., 2022; Lee et al., 2022).

In a nutshell, diffusion models aim to revert a Langevin diffusion chain, moving from a log-concave equilibrium measure $\nu$, say an isotropic Gaussian, back toward a complex, possibly non-log-concave initial measure $\mu$. A Langevin diffusion is a forward chain in the space of probability measures, implemented iteratively with a

stepsize $\eta$ as in (1), where $\nu$ is the equilibrium measure and $\mathbf{f}_k^\mu$ denotes the forward transition map[1] at step $k$.

$$\text{Forward Diffusion} \quad \mu =: \mu_0 \overset{\mathbf{f}_1^\mu}{\to} \mu_\eta \to \cdots \overset{\mathbf{f}_k^\mu}{\to} \mu_{k\eta} \to \cdots \overset{\mathbf{f}_K^\mu}{\to} \mu_{K\eta} \quad \overset{K\to\infty}{\leadsto} \nu \;, \tag{1}$$

$$\text{Time Reversal} \quad \mu =: \mu_0 \underset{\mathbf{b}_1^\mu}{\leftarrow} \mu_\eta \leftarrow \cdots \underset{\mathbf{b}_k^\mu}{\leftarrow} \mu_{k\eta} \leftarrow \cdots \underset{\mathbf{b}_K^\mu}{\leftarrow} \mu_{K\eta} \quad \boxed{\underset{?}{\leftarrow} \cdots \underset{\mathbf{b}^\nu}{\leftarrow} \nu} \;. \tag{2}$$

This forward chain is Markovian and thus time-reversible as in (2), where $\mathbf{b}_k^\mu$ denotes the backward transition map[2] for the $\mu$-chain at step $k$. For sampling, diffusion models propose starting the time reversal process at $K \to \infty$, namely the equilibrium measure $\nu$, and aim to reverse it back to the initial measure $\mu$. However, this is problematic both theoretically (in terms of probability) and conceptually (in terms of optimization). Theoretically, the time reversal of a Markov Chain starting from an equilibrium (invariant measure) will get stuck (Norris, 1998). The backward chain $\nu \leftarrow_{\mathbf{b}^\nu} \nu$ will stay as the invariant measure and never reach $\mu_{K\eta}$. Conceptually, hoping to trace back the initial condition $\mu$ starting from $\nu$ is infeasible. If two forward chains with initials $\mu \neq \mu'$ both end up at $\nu$, the time-reversal starting from $\nu$ (and solely using the future information) cannot recover the past and distinguish these two chains.

Therefore, framing the diffusion model as a time-reversal Markov chain—one that recovers the past from the future—does not fully resolve the underlying conceptual issues. In contrast, we propose studying diffusion models as a form of *sensitivity analysis*. It is clear that starting from $\mu_{K\eta}$ and traversing back following transitions $\mathbf{b}_K^\mu \cdots \mathbf{b}_k^\mu \cdots \mathbf{b}_1^\mu$ will identify $\mu$. But what will happen if we start from a perturbed version $\nu =: \bar{\nu}_{K\eta} \neq \mu_{K\eta}$ and follow the same traversing path $\mathbf{b}_K^\mu \cdots \mathbf{b}_k^\mu \cdots \mathbf{b}_1^\mu$ as illustrated in (4)?

$$\text{Time Reversal} \quad \mu =: \mu_0 \underset{\mathbf{b}_1^\mu}{\leftarrow} \mu_\eta \leftarrow \cdots \underset{\mathbf{b}_k^\mu}{\leftarrow} \mu_{k\eta} \leftarrow \cdots \underset{\mathbf{b}_K^\mu}{\leftarrow} \mu_{K\eta} \quad \boxed{\overset{\times}{\underset{?}{\leftarrow}} \cdots \underset{\mathbf{b}^\nu}{\leftarrow} \nu} \tag{3}$$

$$\text{Backward Denoising} \quad \mu \overset{?}{\leadsto} \bar{\nu}_0 \underset{\mathbf{b}_1^\mu}{\leftarrow} \bar{\nu}_\eta \leftarrow \cdots \underset{\mathbf{b}_k^\mu}{\leftarrow} \bar{\nu}_{k\eta} \leftarrow \cdots \boxed{\underset{\mathbf{b}_K^\mu}{\leftarrow} \bar{\nu}_{K\eta} := \nu} \;, \tag{4}$$

The traversing path carries the past information, namely the signature of the initial measure $\mu$, but starts with an easy-to-sample $\nu$ as a surrogate, replacing $\mu_{K\eta}$. This mismatch renders the backward denoising process (4) *non-Markovian*, thereby making it plausible to recover the past from the future.

**Sensitivity Analysis: Score, Curvature, and Localization.** How can the traversing operator $\mathbf{b}_k^\mu$ be estimated? We show in Proposition 1 that the optimal backward operator $\mathbf{b}_k^\mu$—in terms of transportation cost—depends on the score function $\nabla \log p_{\mu_{k\eta}}$ where $p_{\mu_{k\eta}}$ is the probability density function of $p_{\mu_{k\eta}}$. A key observation in diffusion models is that score estimation can be cast as a supervised prediction problem (Saremi & Hyvärinen, 2019) using the forward diffusion chain, where the goal is to predict the previous location given the current one, in terms of conditional expectation, as seen in Proposition 2.

The fundamental question is whether the perturbation $\bar{\nu}_{K\eta} \approx \mu_{K\eta}$ will get amplified along the backward denoising chain. Will $\bar{\nu}_{k\eta}$ as in (3)-(4) stay close to $\mu_{k\eta}$, for $k = K, \cdots, 0$? This paper provides a precise study of diffusion models from the viewpoint of denoising quality. We unveil in Proposition 3 that the curvature function $\nabla^2 \log p_{\mu_{k\eta}}$ controls the key aspects of this sensitivity analysis. In other words, the curvature function governs the score function's denoising capability, which we refer to as localization. Localization quantifies the uncertainty of the previous location given the current, in terms of conditional covariance.

Most of the current theoretical literature (Lee et al., 2023; Chen et al., 2022; 2023) treats this curvature as a nuisance, for example, by focusing on "non-expansion" metrics $d$, and leveraging a form of data-processing inequality,

$$d(\mu_{k-1}, \bar{\nu}_{k-1})/d(\mu_k, \bar{\nu}_k) \leq 1, \text{ for } d \in \{d_{\text{TV}}, d_{\text{KL}}\} \;.$$

Consequently, $d(\mu_0, \bar{\nu}_0) \leq d(\mu_K, \nu)$, and $d(\mu_0, \bar{\nu}_0)$ can be determined by the contraction of the forward diffusion chain alone. However, even for simple Gaussians, the backward denoising could either be (i) an

---

[1] The rigorous definition will follow in Section 3.

[2] Again, the rigorous definition will follow in Section 3.

expansion or (ii) a contraction much faster than the forward diffusion, under the Wasserstein-2 metric, $W$. Consider the forward diffusion (5) with initialization $\mu \sim \mathcal{N}(0, s^2)$ and temperature $\beta = 1$. Recall, the forward diffusion contracts in $W$ at a rate $1 - \eta$. Our Corollary 1 implies that the one-step backward denoising at effective time $t = k\eta$ with $\mu_k \sim \mathcal{N}(e^{-k\eta}, e^{-2k\eta}s^2 + 1 - e^{-2k\eta})$ satisfies the equality

$$\frac{W(\mu_{k-1}, \bar{\nu}_{k-1})}{W(\mu_k, \bar{\nu}_k)} = 1 + \eta \frac{s^2 - 1}{e^{k\eta} + s^2 - 1} \ , \quad \begin{cases} (i) > 1 & \text{if } s^2 > 1, \\ (ii) < 1 - \eta & \text{if } s^2 < \frac{1}{2}, \text{ and} \\ & k < \frac{1}{2\eta} \log\left(2(1 - s^2)\right). \end{cases}$$

In either case, treating the backward denoising as simply non-expansive under certain metrics is losing significant information about the diffuse-then-denoise process.

In contrast, we study the process under the Wasserstein-2 metric and provide a fine-grained analysis of how a complexity measure based on the curvature, $\nabla^2 \log p_{\mu_{k\eta}}$, completely governs the expansion or contraction of the backward denoising step.

**Diffuse-then-Denoise Process: Offset Contraction/Expansion.** The sensitivity analysis is equivalent to studying the effectiveness of a diffuse-then-denoise process. To simplify the exposition, we consider a one-step version here. Given any two measures, $\mu_0 \neq \nu_0$, run one step of forward diffusion as in (1) with $K = 1$, with $\mathbf{Z}$ isotropic Gaussian and $\beta \in \mathbb{R}_+$, the temperature,

$$\mathbf{X}_\eta = (1 - \eta)\mathbf{X}_0 + \sqrt{2\beta^{-1}\eta}\mathbf{Z}, \ \mathbf{X}_0 \sim \mu_0, \ \text{ and } \ \mathbf{Y}_\eta = (1 - \eta)\mathbf{Y}_0 + \sqrt{2\beta^{-1}\eta}\mathbf{Z}, \ \mathbf{Y}_0 \sim \nu_0 \ .$$

Denote the measure associated with $\mathbf{Y}_\eta \sim \bar{\nu}_\eta$, then run one step of backward denoising with the optimal $\mathbf{b}^\mu$ (as in (4) with $K = 1$) to obtain $\bar{\nu}_0 \leftarrow_{\mathbf{b}^\mu} \bar{\nu}_\eta$. A natural question is whether the cumulative effect of this diffuse-then-denoise process is a contraction. Namely, is $W(\mu_0, \bar{\nu}_0)$ smaller than $W(\mu_0, \nu_0)$?

At first sight, it may seem unnatural that the diffuse-then-denoise process will yield an improvement. How can adding noise and then denoising be better? Consider setting $\beta^{-1} = 0$, and let $\mu_0 = \delta_x$ and $\nu_0 = \delta_y$ be two Dirac measures supported at $x \neq y$. One can verify that

$$\text{Forward } W(\mu_0, \nu_0) = \|x - y\| \ , \ \text{ Backward } W(\mu_\eta, \bar{\nu}_\eta) = (1 - \eta)\|x - y\| \ ,$$
$$\text{Forward-then-Backward } W(\mu_0, \bar{\nu}_0) = (1 - \eta)^{-1} W(\mu_\eta, \bar{\nu}_\eta) = \|x - y\| = W(\mu_0, \nu_0) \ .$$

Here, the backward expansion offsets the forward contraction, resulting in no net effect for the forward-then-backward process.

Curiously, as we shall show in Theorems 1, 2 and 3, roughly speaking, when $\beta^{-1} \neq 0$ and $\nabla^2 \log p_\mu$ has a certain curvature complexity quantified by $\zeta \in \mathbb{R}$, there is a net effect of the diffuse-then-denoise process

$$\text{Forward Diffusion:} \qquad \frac{W(\mu_\eta, \bar{\nu}_\eta)}{W(\mu_0, \nu_0)} \leq 1 - \eta + O(\eta^2) \ ,$$

$$\text{Backward Denoising:} \qquad \frac{W(\mu_0, \bar{\nu}_0)}{W(\mu_\eta, \bar{\nu}_\eta)} \leq 1 + \eta - \eta\beta^{-1}\zeta + O(\eta^2) \ ,$$

$$\text{Diffuse-then-Denoise:} \qquad \frac{W(\mu_0, \bar{\nu}_0)}{W(\mu_0, \nu_0)} \leq 1 - \eta\beta^{-1}\zeta + O(\eta^2) \ .$$

This contrasts sharply with the $\beta^{-1} = 0$ case: curiously, adding non-trivial noise then denoising could be beneficial and result in an effective net contraction $\beta^{-1}\zeta$, provided the curvature $\zeta > 0$. We emphasize that these inequalities become equalities for simple log-concave $\mu_0$'s, thus establishing the sharpness of our characterization, see Corollary 1. In summary, the success or failure of the diffuse-then-denoise process solely depends on the curvature/localization function defined in Proposition 3.

Chaining the argument, we show in Corollary 2 that, rather than log-concavity, it is a multi-scale complexity along all time scales that controls the net effect of the diffuse-then-denoise process (4) with $K$ steps

$$\frac{W(\mu_0, \bar{\nu}_0)}{W(\mu_0, \nu_0)} \leq \exp\left(-\beta^{-1}\eta \sum_{k=1}^{K} \zeta_{k\eta} + O(\eta^2)\right) \ ,$$

where $\zeta_{k\eta}$ is some curvature/localization complexity of $\mu_{k\eta}$ at time $t = k\eta$. For a general initial measure $\mu$, the diffused version $\mu_t$ could be non-log-concave, depending on the time scale $t$. In Section 4, we introduce a notion of multi-scale complexity that describes the localization difficulty for generic $\mu$ and is new to the literature. The multi-scale corresponds to the effective signal-to-noise ratio for the denoising step at each time scale $t$.

**Multi-Scale Complexity: Beyond Log-Concavity.** With any initial measure $\mu$, the Ornstein-Uhlenbeck diffusion process at different time scales $t$ is closely tied to a multi-scale smoothing, at different signal-to-noise ratio $\mathsf{r} = \mathsf{r}(t)$ (defined in (12)),

$$\mathbf{Y}_\mathsf{r} := \mathsf{r}\mathbf{X} + \mathbf{Z} , \ (\mathbf{X}, \mathbf{Z}) \sim \mu \otimes \mathcal{N}(0, 1) .$$

We introduce in Section 4 the following multi-scale complexity, defined based on the localization function $L_\mathsf{r}(y) := \| Cov[\mathsf{r}\mathbf{X}|\mathbf{Y}_\mathsf{r} = y] \|_{op}$,

$$h_\mu(\delta, \mathsf{r}) = \int_{1-\delta}^\infty \mathbb{P}\big(L_\mathsf{r}(\mathbf{Y}_\mathsf{r}) > u\big)\mathrm{d}u , \ \delta \in (0, 1] .$$

Here $\mathbb{P}\big(L_\mathsf{r}(\mathbf{Y}_\mathsf{r}) > u\big)$ is the tail probability of random variable $\| Cov[\mathsf{r}\mathbf{X}|\mathbf{Y}_\mathsf{r}] \|_{op}$, and therefore $h_\mu(\delta, \mathsf{r})$ controls its integrated tail. We show in Theorem 4 that the growth of this integrated tail function $\delta \mapsto h_\mu(\delta, \mathsf{r})$ governs the effectiveness of the diffuse-then-denoise process at any time $t$, with the corresponding SNR scale $\mathsf{r}(t)$. Our characterization holds for any generic $\mu$ that extends far beyond log-concavity.

This multi-scale complexity at every SNR scale $\mathsf{r} \geq 0$, conceptually, is quantifying the curvature $\nabla^2 \log p_{\mathbf{Y}_\mathsf{r}}(y)$ in a certain average sense weighted by $p_{\mathbf{Y}_\mathsf{r}}(y)$. This notion, rather than the worst-case curvature, governs the localization accuracy of the backward denoising chain at every scale. Our analysis is fine-grained in two senses. First, for any $\mathbf{X} \sim \mu$, across different scales of $\mathsf{r}$, the non-log-concavity of $p_{\mathbf{Y}_\mathsf{r}}$ changes in a complex, multi-resolution way. Non-log-concavity across all scales of $\mathsf{r}$ collectively determines the effect of the backward denoising chain. Second, unlike in the worst-case analysis, where the worst point $y$ with the largest positive curvature $\nabla^2 \log p_{\mathbf{Y}_\mathsf{r}}(y)$ dictates the analysis, we leverage the following observation. If a point $y$ with a large positive curvature $\nabla^2 \log p_{\mathbf{Y}_\mathsf{r}}(y)$ is unlikely to occur with $p_{\mathbf{Y}_\mathsf{r}}(y)$ small, the overall non-log-concavity is still benign. This is true in many examples; see Section 5.

This multi-scale complexity may appear mysterious; we present a concrete, non-log-concave example to clarify intuitions. Consider the simplest one-dimensional non-log-concave measure $\mu = \frac{1}{2}\delta_{-1} + \frac{1}{2}\delta_{+1}$.

- Localization and Curvature: $L_\mathsf{r}(y) \leq 1 - \delta$ if and only if $\nabla^2 \log p_{\mathbf{Y}_\mathsf{r}}(y) \preceq -\delta \cdot I_d$, that is, $p_{\mathbf{Y}_\mathsf{r}}(y)$ strongly log-concave at $y$. These $y$'s are good locations with strong curvature: accurate denoising and localization from $y$ is easy. These are locations where diffuse-then-denoise is beneficial. See locations outside the shaded areas in Figure 1.

- Survival Function: provided we have samples $\mathbf{Y}_\mathsf{r}$, the survival function $s_\mathsf{r}(1-\delta) := \mathbb{P}\big(L_\mathsf{r}(\mathbf{Y}_\mathsf{r}) > 1-\delta\big)$ tells us the probability of bad locations with possible non-log-concavity where the backward denoising is hard. It quantifies the mass that may induce a large expansion in the diffuse-then-denoise process. See shaded areas in Figure 1.

- Integrated Tail: slow growth in the integrated tail function $\delta \to h_\mu(\delta, \mathsf{r})$ implies that one can take an effectively large $\delta$ such that the bad locations with positive curvature have a negligible expansion effect, and good locations with strong negative curvature induce a contraction effect, offsetting the expansion. This complexity quantifies an overall notion of curvature.

  Figure 1 shows: (a) low $\mathsf{r} = 0.71$, $h_\mu(0, \mathsf{r}) = h_\mu(0.5, \mathsf{r}) = 0$, no growth, (b) mid $\mathsf{r} = 1.50$, $h_\mu(0, \mathsf{r}) = 0.13, h_\mu(0.5, \mathsf{r}) = 0.24$, rapid growth of integrated tail as $\delta$ increases from 0 to 0.5, and (c) high $\mathsf{r} = 3.00$, $h_\mu(0, \mathsf{r}) = 0.02, h_\mu(0.5, \mathsf{r}) = 0.03$, a very slow growth. Curiously, the complexity is non-monotonic in SNR $\mathsf{r}$: the mid $\mathsf{r}$ presents the hardest non-log-concavity for localization.

How is this multi-scale complexity useful? Here we plot the survival function $s_\mathsf{r}(u)$, indexed by the different SNR $\mathsf{r}$. Curiously, the growth rate of the integrated tail complexity $h_\mu(\delta, \mathsf{r})$ is non-monotonic in SNR $\mathsf{r}$:

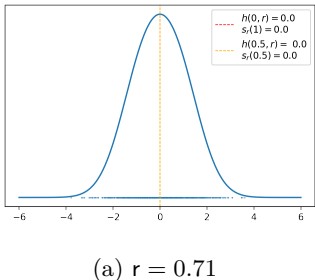 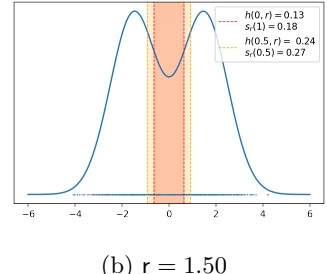 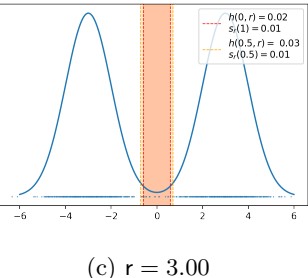

(a) r = 0.71          (b) r = 1.50          (c) r = 3.00

Figure 1: We plot the density $p_{\mathbf{Y}_r}(\cdot)$, for three SNR r's. Red shaded area corresponds to non-log-concave region with $\nabla^2 \log p_{\mathbf{Y}_r}(\cdot) > -\delta$ with $\delta = 0$, and Orange shaded area corresponds to $\delta = 0.5$. For each $\delta$, we report the integrated tail $h_\mu(\delta, r)$ and survival function $s_r(1-\delta)$ for $\delta \in \{0, 0.5\}$. (a) low r = 0.71, $s_r(1) = 0$, $s_r(0.5) = 0$; (b) mid r = 1.50, $s_r(1) = 0.18$, $s_r(0.5) = 0.27$, non-trivial mass of bad locations; (c) high r = 3.00, $s_r(1) = 0.01$, $s_r(0.5) = 0.01$, though bad locations do exist, samples $\mathbf{Y}_r$ rarely end up there.

both low SNR r $\leq$ 1 and high r $\geq$ 2 have extremely slow growth in the integrated tail; the hardest non-log-concavity happens when r $\in (1, 2)$. Conceptually, for any given initial measure $\mu$, the multi-scale complexity tells us precisely at what time scale r(t) the backward denoising chain suffers the most. See Section 5 for more comprehensive examples.

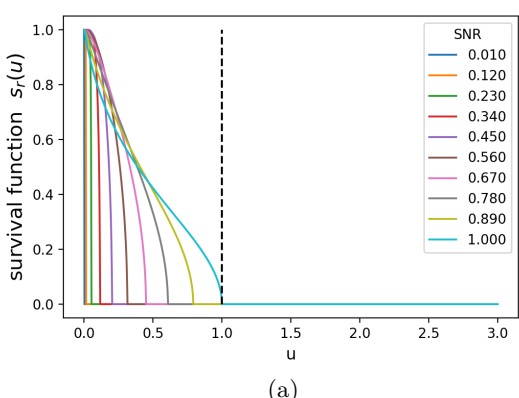 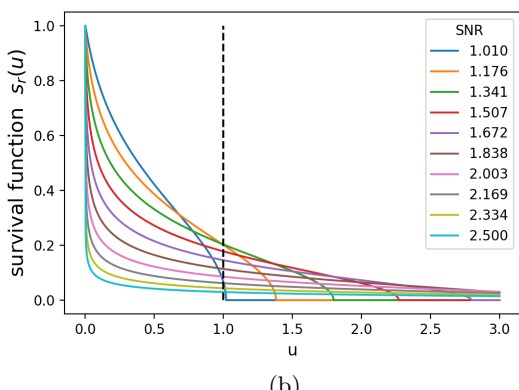

(a)                                   (b)

Figure 2: (a) $s_r(u)$ Low SNR. (b) $s_r(u)$ High SNR.

## 1.1 Preliminaries and Notations

**Notation** Throughout this paper, we consider $\mu \in \mathcal{P}_2(X)$, the space of probability measures with a bounded second moment with $X \subseteq \mathbb{R}^d$. $\mathcal{P}_2^r(X) \subset \mathcal{P}_2(X)$ denotes probability measures absolutely continuous to the Lebesgue measure. For a measure $\nu \in \mathcal{P}_2^r(X)$, denote $\nu = p_\nu \cdot \mathcal{L}^d$ the Radon-Nikodym derivative w.r.t Lebesgue measure, where $p_\nu \in C^1(X)$ is the density function. We reserve $\mathcal{G}, \mathcal{F}, \mathcal{E}$ for functionals $\mathcal{P}_2(X) \to \mathbb{R}$, and $h, g, f$ for real-valued functions $X \to \mathbb{R}$. Let $\boldsymbol{\xi} \in C_c^\infty(X; X)$ denote smooth vector fields; $\mathbf{i}$ denotes the identity map, $\mathbf{t}, \mathbf{f}, \mathbf{b}$ denotes the optimal transport maps, in Definition 2. We use $\mathbf{X}, \mathbf{Y}, \mathbf{Z}, \mathbf{B}$ to denote random vectors. For a matrix $M$, $\|M\|_{op}$ denotes the operator norm, $tr(M)$ denotes the trace of the matrix. For a vector $v$, $\|v\|$ denotes the Euclidean norm.

The forward Langevin diffusion process is a stochastic differential equation in the form,

$$\mathrm{d}\mathbf{X}_t = -\nabla f(\mathbf{X}_t)\mathrm{d}t + \sqrt{2\beta^{-1}}\mathrm{d}\mathbf{B}_t, \quad \nabla f \in C_c^\infty(X, X). \tag{5}$$

Here $\beta^{-1}$ serves as the temperature parameter, which controls the magnitude of the injected noise $\mathrm{d}\mathbf{B}_t$. The probability measure of $\mathbf{X}_t$, denoted as $\mu_t$ with density $\rho_t := p_{\mu_t}$, evolves according to the Fokker-Planck partial differential equation

$$\partial_t \rho_t = \nabla \cdot \left( \rho_t (\nabla f + \beta^{-1} \nabla \log \rho_t) \right) . \tag{6}$$

Forward diffusion can also be viewed as a gradient flow in the Wasserstein space. For two measures $\mu, \nu \in \mathcal{P}_2^r(X)$, define the Wasserstein metric as

$$W_2(\mu, \nu) := \min_{\pi \in \Pi(\mu, \nu)} \left( \int \|x - y\|^2 \mathrm{d}\pi(x, y) \right)^{1/2} .$$

Jordan et al. (1998) showed that the forward Fokker-Planck PDE, $\mu_t \to \mu_{t+\eta}$, can be viewed as the steepest descent with respect to the Wasserstein metric in the infinitesimal limit, as $\eta \to 0$

$$\mu_{t+\eta} := \underset{\nu \in \mathcal{P}_2^r(X)}{\arg\min} \frac{1}{2\eta} W_2^2(\mu_t, \nu) + \mathcal{G}(\nu), \ \mathcal{G} : \mathcal{P}_2^r(X) \to \mathbb{R} . \tag{7}$$

Here the functional $\mathcal{G}(\nu) = \mathcal{F}(\nu) + \beta^{-1}\mathcal{E}(\nu)$ consists of two parts, a potential functional $\mathcal{F}$ and an entropy functional $\mathcal{E}$

$$\mathcal{F}(\nu) := \int f \, \mathrm{d}\nu = \int f(x) p_\nu(x) \, \mathrm{d}x \ ,$$

$$\mathcal{E}(\nu) := \int \log\left(\frac{\mathrm{d}\nu}{\mathrm{d}x}\right) \mathrm{d}\nu = \int p_\nu(x) \log p_\nu(x) \, \mathrm{d}x \ .$$

## 1.2 Related Work

Probabilistic generative models, including generative adversarial networks, flow-based generative models, and diffusion-based generative models, have recently received broad research interest, both empirically (Goodfellow et al., 2020; Song & Ermon, 2019; 2020; Song et al., 2021; Karras et al., 2022; Nichol & Dhariwal, 2021; Kingma et al., 2021; Huang et al., 2023) and theoretically (Papamakarios et al., 2021; Liang, 2021; Hur et al., 2024; Oko et al., 2023; Lübeck et al., 2022; Chen et al., 2024; Block et al., 2020; DeBortoli et al., 2021; Lee et al., 2022; Montanari, 2023). Generative models with diffusion-based sampling can be traced back to three influential formulations: Denoising Diffusion Probabilistic Models (DDPMs) (Ho et al., 2020; Sohl-Dickstein et al., 2015), Score-based Generative Models (SGMs) (Song & Ermon, 2019), and Score-based Stochastic Differential Equations (Song et al., 2021; Karras et al., 2022). The latter universalizes the frameworks by discretizing a particular Stochastic Differential Equation at specific signal-to-noise ratios (SNR). Sampling can be viewed as Langevin dynamics for the time-reversed SDE (Anderson, 1982; Ho et al., 2020; Dockhorn et al., 2021), or deterministic transports via a probability flow ODE (Song et al., 2020; Maoutsa et al., 2020; Guo et al., 2022). The deterministic denoising outlined in Song et al. (2020) is optimal in terms of transportation of measure in the Wasserstein metric. This was previously noted by Chen et al. (2024); Jordan et al. (1998) in the infinitesimal limit $\eta \to 0$. We show in Proposition 1 that for a fixed stepsize $\eta$, the score denoising map is the optimal transport map in the backward chain.

Rigorous justification for diffuse-then-denoise can be found in the seminal work of Saremi & Hyvärinen (2019) who unified two distinct schemes: (1) smoothing via Gaussian convolution $\mathbf{Y} = \mathbf{X} + \frac{1}{r}\mathcal{N}(0, I_d)$, and (2) denoising via the score function $\mathbb{E}[\mathbf{X}|\mathbf{Y} = y] = y + \frac{1}{r^2}\nabla \log p_\mathbf{Y}(y)$, to design a single machine for sampling $\mathbf{X}$. Their approach is motivated by a fundamental result in the concentration of measure literature (Vershynin, 2018); that $d$-dimensional Gaussian vectors concentrate on a uniform sphere of radius $\sqrt{d}$ in high-dimensions. And so, while $\mathbf{X}$ may be non-log-concave or confined to a low-dimensional manifold, $\mathbf{Y}$ may not be. Thus, diffuse ($\mathbf{X} \to \mathbf{Y}$) then denoise ($\mathbf{Y} \to \mathbf{X}$) presents a way to sample when the distribution of $\mathbf{X}$ is misbehaved. Exploiting the machinery of measure transportation, we show that Gaussian convolution up to an appropriate signal-to-noise ratio $r$ promotes log-concavity, yielding desirable contractive properties for denoising, $\mathbf{Y} \to \mathbf{X}$.

Recent empirical work (Song & Ermon, 2019; 2020; Song et al., 2021; Karras et al., 2022) demonstrates dramatic improvements in sample quality and log-likelihood metrics for diffusions by employing a rich scheme of noise scheduling, namely the effective SNR $t \mapsto r(t)$, a map between time and the corresponding SNR. Several authors consider jointly learning the schedule alongside diffusion network parameters (Nichol & Dhariwal, 2021; Kingma et al., 2021). While state-of-the-art applications suggest that convolving at multiple noise scales is advantageous (Dhariwal & Nichol, 2021; Austin et al., 2021), its theoretical benefit remains to be understood. Restricting to the Ornstein-Uhlenbeck process, we discover a multi-scale complexity measure that controls the effective contraction of diffuse-then-denoise at each SNR $r$. Particularly for non-log-concave distributions, a wide regime of SNR schedule $r(t)$ may be necessary as diffusing at a specific SNR scale $r$ pushes the problem into an effective near-log-concave setting with strong curvature. Our multi-scale complexity determines the effective curvature, thereby providing a vehicle for probing the noise-scheduling question.

Theoretical investigation of diffusion-based sampling mostly focuses on the non-expansive properties of the denoising process under $f$-divergence, say total variation $d_{\mathrm{TV}}$ or Kullback-Leibler $d_{\mathrm{KL}}$, where data processing inequities hold. The current theory falls short in addressing the behavior of the backward denoising process under the Wasserstein metric, where the backward denoising chain does incur expansion even for simple Gaussian distributions (Chen et al., 2022). Typically, the problem is coupled with the additional burden of having access to only an estimated score function (Block et al., 2020; DeBortoli et al., 2021; Lee et al., 2022). This presents a significant challenge; however, it is not the focus of the current paper. We isolate treatment of the backward denoising question: initializing at $\mu_T$, and the equilibrium measure, $\nu$, suppose we observe $\{\mu_0 \leftarrow_{\mathbf{b}_1^\mu} \ldots \leftarrow_{\mathbf{b}_K^\mu} \mu_K\}$, $\{\bar{\nu}_0 \leftarrow_{\mathbf{b}_1^\mu} \ldots \leftarrow_{\mathbf{b}_K^\mu} \bar{\nu}_K := \nu\}$. How does $d(\mu_K, \bar{\nu}_K) \to d(\mu_{K-1}, \bar{\nu}_{K-1}) \to \cdots \to d(\mu_0, \bar{\nu}_0)$ evolve? Is the backward denoising chain expansive or contractive at each time scale $t = k\eta$? What geometric complexities govern the quality of the denoising at each time scale? We view our focus as complementary to the current literature, which conducts careful sensitivity analyses of score estimation and time discretization.

Provided the score functions are accurate in an $L_2$ sense, Block et al. (2020); Lee et al. (2022) established results in the Wasserstein metric for unimodal distributions, for example, those satisfying strong dissipativity or a log-Sobolev inequality. These results are extended in Lee et al. (2023); Chen et al. (2022; 2023) to allow substantial non-log-concavity for reverse SDE sampling, and in Chen et al. (2024) for the probability flow ODE. However, these analyses are conducted in $d_{\mathrm{TV}}$ or $d_{\mathrm{KL}}$ and avoid the detailed curvature information in the backward map. For $d_{\mathrm{TV}}$ or $d_{\mathrm{KL}}$, and a finite $K$, one can always appeal to a data processing inequality to bound $d(\mu_0, \bar{\nu}_0) \leq d(\mu_K, \nu)$, solely determined by the forward diffusion chain. This is not fine-grained enough to understand denoising. It overlooks the curvature information and the amount of non-log-concavity at any specific SNR $r$. As in the introduction, the backward denoising step could expand significantly or contract much faster than the forward diffusion under the Wasserstein metric.

More recently, Bruno et al. (2023); Gao et al. (2025) establish results in the Wasserstein metric under log-concavity assumptions. The concurrent work (Gentiloni-Silveri & Ocello, 2025) tackles the problem assuming weak log-concavity of the target density, a property with origins in dissipativity - that the target density is essentially log-concave outside a region of compact support. Our work makes no such assumptions. Similar to Gentiloni-Silveri & Ocello (2025), we note that the denoising process can exhibit non-contractive behavior. In our analysis, the fine-grained behavior of the denoising process is governed by a notion of average curvature, defined as an integrated tail function measuring the relative mass of locations with positive curvature versus those with negative curvature. Notions of average curvature have been used to derive improved convergence bounds for diffusion models with respect to $d_{\mathrm{KL}}$ (Chen et al., 2023; Benton et al., 2023). Our paper shows that a different notion of average curvature also plays a role in the sensitivity analysis when the metric is the Wasserstein distance. Our perspective helps bridge the parallel theoretical analyses of diffusion models under $f$-divergence and Wasserstein distance.

More importantly, previous approaches disconnect denoising from diffusion when evaluating the success of score-based sampling. We show that the net benefit of the diffuse-then-denoise process matters: the contraction of the forward diffusion, offset by the possible backward expansion, results in a net benefit as long as there is enough negative curvature on average. This paradigm shift is enabled by a new notion of

multi-scale complexity governing the effective curvature and "localization" effect of the diffuse-then-denoise process. This allows an analysis that extends beyond log-concavity assumptions.

## 2 Denoising and Localization: Score and Curvature

### 2.1 Score Function: Denoising and Optimal Transport

The score function arises naturally in the Fokker-Planck PDE (6). For a valid probability density $\rho$, the vector field $\nabla \log \rho : X \to \mathbb{R}^d$ defines the score function.

**Definition 1** (Score Function). *For a measure $\nu \in \mathcal{P}_2^r(X)$, denote its density with respect to the Lebesgue measure as $p_\nu$ where $\nu = p_\nu \cdot \mathcal{L}^d$. The score function is $x \mapsto \nabla \log p_\nu(x)$.*

It turns out, even for non-vanishing $\eta$, the score function $\nabla \log \rho_{t+\eta}$ induces an optimal plan to localize and denoise from $\mu_{t+\eta}$, defined in (7), back to $\mu_t$, in the sense of optimal transport (OT). In the $\eta \to 0$ case, the score induced by the OT map was studied in Chen et al. (2024); Song et al. (2020); Jordan et al. (1998). We first introduce the OT map.

**Definition 2** (OT Map, (Brenier, 1987)). *For $\mu, \nu \in \mathcal{P}_2^r(X)$, there exists a unique optimal transport map $\mathbf{t}_\nu^\mu : X \to X$ that solves the Monge problem*

$$\mathbf{t}_\nu^\mu = \underset{\mathbf{t}:\mathbf{t}_\#\nu=\mu}{\arg\min} \left( \int \|y - \mathbf{t}(y)\|^2 \, \mathrm{d}\nu(y) \right)^{1/2} .$$

*and attains the minimum of the Kantorovich problem*

$$W_2(\mu, \nu) = \left( \int \|\mathbf{i} - \mathbf{t}_\nu^\mu\|^2 \mathrm{d}\nu \right)^{1/2} .$$

**Proposition 1** (Score Function and Backward OT Map). *Consider the Wasserstein gradient descent as in (7) with a lower-semicontinuous $\mathcal{G} = \mathcal{F} + \beta^{-1}\mathcal{E}$. Assume that there exists $\eta_\star > 0$, such that for all $\eta \in (0, \eta_\star)$, $\mu_{t+\eta}$ in (7) admits a well-defined minimizer. Then for any $\eta \in (0, \eta_\star)$, the optimal transport map $\mathbf{t}_{\mu_{t+\eta}}^{\mu_t}$ as in Definition 2 takes the form,*

$$\frac{1}{\eta}(\mathbf{t}_{\mu_{t+\eta}}^{\mu_t} - \mathbf{i})(x) = \nabla f(x) + \beta^{-1}\nabla \log p_{\mu_{t+\eta}}(x), \text{ for } \mu_{t+\eta}\text{-a.e. } x \in \mathbb{R}^d .$$

Recall that discretized Langevin diffusion reads $\mathbf{X}_{t+\eta} = \mathbf{X}_t - \eta\nabla f(\mathbf{X}_t) + \sqrt{2\beta^{-1}\eta}\mathbf{Z}$. Another interpretation of the score function is that it induces a backward denoising step for the diffusion, namely, quantifying the barycenter $\mathbb{E}[\mathbf{X}_t - \eta\nabla f(\mathbf{X}_t) \mid \mathbf{X}_{t+\eta} = y]$. In other words, score estimation can be cast as a prediction problem based on the diffusion process, where one aims to predict $\mathbf{X}_t - \eta\nabla f(\mathbf{X}_t)$ based on $\mathbf{X}_{t+\eta}$ (Saremi & Hyvärinen, 2019).

**Proposition 2** (Score and Backward Denoising). *Consider $\mathbf{Y} = \mathbf{X} + \sigma\mathbf{Z}$, where $\mathbf{X} \sim \mu$ and $\mathbf{Z} \sim \mathcal{N}(0, I_d)$ and $\mathbf{X}, \mathbf{Z}$ are independent. Let $p_\mathbf{Y}$ denote the density function associated with the random variables $\mathbf{Y}$. Then*

$$\nabla \log p_\mathbf{Y}(y) = -\frac{1}{\sigma^2}\left\{ y - \mathbb{E}[\mathbf{X}|\mathbf{Y} = y] \right\} .$$

The score function is the optimal transport map to denoise the diffusion process, but how accurate is the denoising step? Conceptually, the denoising quality depends on the localization $Cov[\mathbf{X}_t - \eta\nabla f(\mathbf{X}_t) \mid \mathbf{X}_{t+\eta} = y]$. As we shall see next, the localization quality of the score function as a backward denoising step depends on the curvature function, $x \mapsto \nabla^2 \log p_\nu(x)$.

### 2.2 Curvature Function: Backward Localization

The curvature function, namely, the derivative of the score function, governs whether the backward denoising step is localized. The following proposition describes the variability of the backward denoising, $Cov[\mathbf{X}_t - \eta\nabla f(\mathbf{X}_t) \mid \mathbf{X}_{t+\eta} = y]$. Intuitively, a large positive curvature of the log density function results in a large conditional covariance and in turn, makes the denoising process hard.

**Proposition 3** (Curvature and Localization). *Consider $\mathbf{Y} = \mathbf{X} + \sigma\mathbf{Z}$, where $\mathbf{X} \sim \mu$ and $\mathbf{Z} \sim \mathcal{N}(0, I_d)$ and $\mathbf{X}, \mathbf{Z}$ are independent. Let $p_{\mathbf{Y}}$ denote the density function associated with the random variables $\mathbf{Y}$. Then*

$$\nabla^2 \log p_{\mathbf{Y}}(y) = -\frac{1}{\sigma^2}\Big\{I_d - Cov[\tfrac{\mathbf{X}}{\sigma}|\mathbf{Y} = y]\Big\} \, .$$

The validity of the diffusion-then-denoise process depends on the spectrum of the conditional covariance function. Motivated by this, we shall define a multi-scale complexity measure in Section 4. Before concluding this section, we show that the average-case curvature is always negative, although the worst-case curvature is positive for non-log-concave measures. In other words, average curvature $tr[\nabla^2 \log p_\nu(y)]$ weighed by $p_\nu(y)$, is strictly negative for any $\nu$. This will be useful later.

**Proposition 4** (Curvature and Score).

$$\int tr[\nabla^2 \log p_\nu(x)]\mathrm{d}\nu(x) = -\int \|\nabla \log p_\nu(x)\|^2 \mathrm{d}\nu(x) \, .$$

An immediate implication is that the average localization radius is bounded.

$$\mathbb{E}\, \| Cov[\tfrac{\mathbf{X}}{\sigma}|\mathbf{Y}]\|_{op} \leq \mathbb{E}\, tr\,\big[\, Cov[\tfrac{\mathbf{X}}{\sigma}|\mathbf{Y}]\,\big] \leq d \, .$$

As we shall see in Definition 4 in Section 4.2, the tail behavior of $\| Cov[\tfrac{\mathbf{X}}{\sigma}|\mathbf{Y}]\|_{op}$ governs the complexity of the diffuse-then-denoise process.

We conclude this section by noting that the score function $\nabla \log p(y)$, the curvature function $\nabla^2 \log p(y)$, and higher-order generalizations play a central role in characterizing the quality of distributional denoising. In particular, Liang (2025a) analyzes the universality of distributional denoisers constructed from score and curvature functions, while Liang (2025b) proposes a hierarchy of denoisers based on higher-order scores that, in the limit, converges to the optimal map transporting $P_{\mathbf{Y}}$ to $P_{\mathbf{X}}$.

## 3 Forward Contraction and Backward Expansion

In this section, we study how the curvature governs the effectiveness of the diffuse-then-denoise process in diffusion models.

### 3.1 Contraction of Forward Chain

Consider two forward diffusion chains $\mu_t, \nu_t$ at a given time $t$, with an infinitesimal time increment $\eta$. Recall the Fokker-Planck PDE in (6); the measures are implemented by

$$
\begin{aligned}
\mu_{t+\eta} &:= \mathbf{f}^{\mu,\eta}_{\#}\mu_t, \text{ where } \mathbf{f}^{\mu,\eta} = \mathbf{i} - \eta(\nabla f + \beta^{-1}\nabla \log p_{\mu_t}) \, , \\
\nu_{t+\eta} &:= \mathbf{f}^{\nu,\eta}_{\#}\nu_t, \text{ where } \mathbf{f}^{\nu,\eta} = \mathbf{i} - \eta(\nabla f + \beta^{-1}\nabla \log p_{\nu_t}) \, .
\end{aligned}
\tag{8}
$$

Again, these are the Euler discretization of the Wasserstein gradient flow as in (7), with functional $\mathcal{G}(\nu) = \mathcal{F}(\nu) + \beta^{-1}\mathcal{E}(\nu)$.

**Theorem 1** (Forward Contraction). *Assume for some $\lambda \in \mathbb{R}_+$, $\nabla^2 f(x) \succeq \lambda \cdot I_d$, $\forall x \in X$. For any $\mu_t \neq \nu_t \in \mathcal{P}_2^r(X)$ and any $\beta \in \mathbb{R}_+$, the one-step forward process (8) satisfies*

$$\limsup_{\eta \to 0} \frac{1}{\eta}\frac{W_2^2(\mu_{t+\eta}, \nu_{t+\eta}) - W_2^2(\mu_t, \nu_t)}{W_2^2(\mu_t, \nu_t)} \leq -2\lambda \, .$$

*Remark* 1. The above Theorem 1 shows that as $\eta \to 0$, for any $\mu_t, \nu_t$

$$\frac{W_2(\mu_{t+\eta}, \nu_{t+\eta})}{W_2(\mu_t, \nu_t)} \leq 1 - \eta\lambda \, .$$

With the choice $\nu_t = \nu_\star$, the invariant measure, we have $\nu_{t+\eta} = \nu_\star$, and thus

$$\frac{W_2(\mu_{t+\eta}, \nu_\star)}{W_2(\mu_t, \nu_\star)} \leq 1 - \eta\lambda \ .$$

Conceptually, the one-step contraction rate for the forward diffusion process is governed by $\lambda$. This contraction rate is sharp and holds equality for some $\mu, \nu$'s, for example, take $\mu, \nu$ as Dirac measures.

## 3.2 Expansion of Backward Chain

Recall the backward OT map as in Proposition 1

$$\mathbf{b}^{\mu,\eta} = \mathbf{i} + \eta(\nabla f + \beta^{-1}\nabla \log p_{\mu_t}) \ . \tag{9}$$

This backward OT map encapsulates the plan to revert the chain $\mathbf{b}_{\#}^{\mu,\eta}\mu_t \to \mu_{t-\eta}$, which we will formally prove in Theorem 3 in next section. In this section, we conduct a sensitivity analysis on the backward map: consider a measure $\nu$ that is a small perturbation to $\mu_t$, will $\mathbf{b}_{\#}^{\mu,\eta}\nu$ stay close to $\mathbf{b}_{\#}^{\mu,\eta}\mu_t$? This question concerns the expansion rate of the backward chain.

We first derive a warm-up result in the simple case, employing a notion of worst-case curvature. Later, we will generalize the result in Section 4, elucidating how an average-case curvature gives rise to a multi-resolution complexity measure that governs the effectiveness of the diffuse-then-denoise process.

**Theorem 2** (Backward Expansion). *Consider $\nabla^2 f(x), \nabla^2 \log p_{\mu_t}(x)$ with bounded eigenvalues over $x \in X$. Assume for some $\kappa \in \mathbb{R}_+$, $\nabla^2 f(x) \preceq \kappa \cdot I_d$, $\forall x \in X$. Assume $\nabla \log p_{\mu_t} \in C_c^\infty(X, X)$ and for some $\zeta \in \mathbb{R}$*

$$\nabla^2 \log p_{\mu_t}(x) \preceq -\zeta \cdot I_d, \ \forall x \in X \ .$$

*Then for any $\beta \in \mathbb{R}_+$, the one-step backward map (9) satisfies*

$$\limsup_{\eta \to 0} \sup_{\nu \in \mathcal{P}_2^r(X)} \frac{1}{\eta} \frac{W_2^2(\mathbf{b}_{\#}^{\mu,\eta}\mu_t, \mathbf{b}_{\#}^{\mu,\eta}\nu) - W_2^2(\mu_t, \nu)}{W_2^2(\mu_t, \nu)} \leq 2(\kappa - \beta^{-1}\zeta) \ .$$

*Remark* 2. We emphasize that this theorem operates even when $\zeta < 0$, namely when $\mu_t$ is non-log-concave. The above theorem shows that, for any $\nu \in \mathcal{P}_2^r(X)$, as $\eta \to 0$

$$\frac{W_2(\mathbf{b}_{\#}^{\mu,\eta}\mu_t, \mathbf{b}_{\#}^{\mu,\eta}\nu)}{W_2(\mu_t, \nu)} \leq 1 + \eta(\kappa - \beta^{-1}\zeta) \ .$$

Conceptually, the one-step expansion rate for the backward OT map is governed by $\kappa - \beta^{-1}\zeta$. We call this an expansion because it is positive when $\beta$ is large enough, regardless of the sign of $\zeta$.

This backward expansion is inevitable, even for certain log-concave $\mu_t$. As a simple corollary of Theorem 2, we show the expansion upper bound is tight.

**Corollary 1** (Lower Bound). *Assume for some $\kappa, \zeta \in \mathbb{R}_+$,*

$$\nabla^2 f(x) = \kappa \cdot I_d, \ \nabla^2 \log p_{\mu_t}(x) = -\zeta \cdot I_d \ .$$

*Then for any $\beta > \zeta/\kappa$, the one-step backward map (9) satisfies for all $\nu$*

$$\lim_{\eta \to 0} \frac{1}{\eta} \frac{W_2^2(\mathbf{b}_{\#}^{\mu,\eta}\mu_t, \mathbf{b}_{\#}^{\mu,\eta}\nu) - W_2^2(\mu_t, \nu)}{W_2^2(\mu_t, \nu)} = 2(\kappa - \beta^{-1}\zeta) > 0 \ .$$

## 3.3 Diffuse-then-Denoise: One-Step Improvement and Chaining

Consider the special case of an Ornstein-Uhlenbeck process where $f(x) = \|x\|^2/2$. Take any two chains $\mu_t$ and $\nu_t$, as $\eta \to 0$, then Theorem 1 claims the forward diffusion satisfies contraction

$$\frac{W_2(\mathbf{f}_{\#}^{\mu,\eta}\mu_t, \mathbf{f}_{\#}^{\nu,\eta}\nu_t)}{W_2(\mu_t, \nu_t)} = \frac{W_2(\mu_{t+\eta}, \nu_{t+\eta})}{W_2(\mu_t, \nu_t)} \leq 1 - \eta + O(\eta^2) \ .$$

The backward denoising in Theorem 2 satisfies an expansion at most

$$\frac{W_2(\mathbf{b}_\#^{\mu,\eta}\mathbf{f}_\#^{\mu,\eta}\mu_t, \mathbf{b}_\#^{\mu,\eta}\mathbf{f}_\#^{\nu,\eta}\nu_t)}{W_2(\mathbf{f}_\#^{\mu,\eta}\mu_t, \mathbf{f}_\#^{\nu,\eta}\nu_t)} = \frac{W_2(\mathbf{b}_\#^{\mu,\eta}\mu_{t+\eta}, \mathbf{b}_\#^{\mu,\eta}\nu_{t+\eta})}{W_2(\mu_{t+\eta}, \nu_{t+\eta})} \leq 1 + \eta(1 - \beta^{-1}\zeta) + O(\eta^2) \ .$$

We shall show next in Theorem 3 that

$$W_2(\mathbf{b}_\#^{\mu,\eta}\mathbf{f}_\#^{\mu,\eta}\mu_t, \mu_t) = O(\eta^2) \ .$$

Then compared to $\nu_t$, the diffuse-then-denoise version $\mathbf{b}_\#^{\mu,\eta}\mathbf{f}_\#^{\nu,\eta}\nu_t$ will get closer to $\mu_t$, in the following sense

$$\frac{W_2(\mu_t, \mathbf{b}_\#^{\mu,\eta}\mathbf{f}_\#^{\nu,\eta}\nu_t)}{W_2(\mu_t, \nu_t)} \leq 1 - \eta\beta^{-1}\zeta + O(\eta^2) \ .$$

The one-step diffuse-then-denoise process $\mathbf{b}^{\mu,\eta} \circ \mathbf{f}^{\nu,\eta}$ has a net contraction $\beta^{-1}\zeta$ when $\zeta > 0$, namely, the forward contraction, offset by the possible backward expansion, presents a net gain in localization.

**Theorem 3.** *Define the one-step diffuse-then-denoise*

$$\mathbf{f}^{\mu,\eta} := \mathbf{i} - \eta(\nabla f + \beta^{-1}\nabla\log p_\mu), \ \ \mu_\eta := \mathbf{f}_\#^{\mu,\eta}\mu \ ,$$
$$\mathbf{b}^{\mu,\eta} := \mathbf{i} + \eta(\nabla f + \beta^{-1}\nabla\log p_{\mu_\eta}) \ .$$

*Assume $\nabla f, \nabla\log p_\mu, \nabla\log p_{\mu_\eta} \in C_c^\infty(X, X)$, then*

$$W_2(\mathbf{b}_\#^{\mu,\eta}\mathbf{f}_\#^{\mu,\eta}\mu, \mu) = O(\eta^2) \ .$$

A direct corollary concerning the diffuse-then-denoise chain now follows from Theorem 1, 2, and 3. We consider finite $K$-step diffuse-then-denoise chains:

- Forward diffuse: for the $\mu$ chain, set $\mu_0 = \mu$, and define $\mathbf{f}_k^{\mu,\eta} := \mathbf{i} - \eta(\nabla f + \beta^{-1}\nabla\log p_{\mu_{(k-1)\eta}})$ and $\mu_{k\eta} := (\mathbf{f}_k^{\mu,\eta})_\#\mu_{(k-1)\eta}$, recursively for $k = 1, 2, \ldots, K$. Same for the $\nu$ chain.

- Backward denoise: for the $\nu$ chain, recursively apply the backward OT map $\mathbf{b}_k^{\mu,\eta} := \mathbf{i} + \eta(\nabla f + \beta^{-1}\nabla\log p_{\mu_{k\eta}})$ to $\nu_{K\eta}$, for $k = K, K-1, \ldots, 1$.

**Corollary 2** (Diffuse-then-Denoise: Chaining)**.** *Let $f(x) = \|x\|^2/2$. Define the diffuse-then-denoise map*

$$\mathbf{f}_{[K]}^\nu := \mathbf{f}_K^{\nu,\eta} \circ \cdots \circ \mathbf{f}_2^{\nu,\eta} \circ \mathbf{f}_1^{\nu,\eta} \ ,$$
$$\mathbf{b}_{[K]}^\mu := \mathbf{b}_1^{\mu,\eta} \circ \mathbf{b}_2^{\mu,\eta} \circ \cdots \circ \mathbf{b}_K^{\mu,\eta} \ .$$

*Assume there exists a sequence of $\zeta_{k\eta} \in \mathbb{R}$ such that*

$$\nabla^2 \log p_{\mu_{k\eta}}(x) \preceq -\zeta_{k\eta} \cdot I_d, \ \forall x \in X, \ \forall k = 1, 2, \ldots, K \ .$$

*Then the diffuse-then-denoise process on $\nu$ with fixed $K$ steps, denoted as $\left(\mathbf{b}_{[K]}^\mu \circ \mathbf{f}_{[K]}^\nu\right)_\#\nu$, satisfies*

$$\frac{W_2^2\left(\mu, \left(\mathbf{b}_{[K]}^\mu \circ \mathbf{f}_{[K]}^\nu\right)_\#\nu\right)}{W_2^2(\mu, \nu)} \leq \exp\left(-2\eta\beta^{-1}\sum_{k=1}^K \zeta_{k\eta} + O(\eta^2)\right) \ . \tag{10}$$

The above Corollary states the denoising quality of the diffuse-then-denoise chain is collectively determined by the curvature at each step $\zeta_{k\eta}$. For non-log-concave $\mu$, along the chain, some of the $\zeta_{k\eta}$ will be positive, and some will be negative. Therefore, we need a fine-grained understanding of the curvature at each time scale $t = k\eta$ to understand the whole chain behavior. This calls for a fine-resolution analysis of the curvature of $\mu_t$, going beyond the worst-case curvature, the focus of the next section.

# 4 Beyond Log-Concavity: A Multi-Scale Complexity

In this section, we restrict to the Ornstein-Uhlenbeck process and discover a multi-scale complexity measure that controls the effective contraction/expansion of the diffuse-then-denoise process. The key is that for the Ornstein-Uhlenbeck process, $\mu_t$ is a smoothed version of $\mu$ at a particular signal-to-noise ratio scale $\mathsf{r} = \mathsf{r}(t)$.

## 4.1 OU Process

**Definition 3** (Ornstein-Uhlenbeck Process). *Define the Ornstein-Uhlenbeck Process with initialization $X_0 \sim \mu$, and potential $f(x) = \|x\|^2/2$*

$$\mathrm{d}\mathbf{X}_t = -\mathbf{X}_t \mathrm{d}t + \sqrt{2\beta^{-1}}\mathrm{d}\mathbf{B}_t \ .$$

*Then the distribution of $\mathbf{X}_t, \forall t \in \mathbb{R}_+$ admits the representation*

$$\mathbf{X}_t \stackrel{\mathcal{L}}{\sim} e^{-t}\mathbf{X}_0 + \sqrt{\beta^{-1}(1 - e^{-2t})}\mathbf{Z}, \ \mathbf{Z} \sim \mathcal{N}(0, I_d) \ . \tag{11}$$

For any measure $\mathbf{X} \sim \mu \in \mathcal{P}_2(X)$, the above representation motivates us to consider a sequence of problems indexed by a multi-scale signal-to-noise ratio. Recall Proposition 3,

$$\mathsf{r}(t) := \frac{e^{-t}}{\sqrt{\beta^{-1}(1 - e^{-2t})}}, \ \mathsf{s}(t) := \sqrt{\beta^{-1}(1 - e^{-2t})} \ ,$$

$$\nabla^2 \log p_{\mu_t}(x) = -\frac{1}{\mathsf{s}^2(t)}\left\{I_d - Cov[\mathsf{r}(t)\mathbf{X}_0 | \mathbf{X}_t = x]\right\} \ . \tag{12}$$

The curvature at each scale quantifies and, collectively, determines the measure's complexity.

If $\mu$ is a log-concave measure, then by Prékopa-Leindler Inequality, $\mu_t$'s are convolutions of a log-concave measure with a Gaussian measure, and therefore, log-concave. Hence, for any $t$, we know that $\nabla^2 \log p_{\mu_t}(x) \preceq 0$. As a direct consequence of Equation (10), we know that the diffuse-then-denoise chain is effective.

However, for non-log-concave measure $\mu$, the $\nabla^2 \log p_{\mu_t}(x)$ may have positive eigenvalues. Given Equation (12), studying the positive eigenvalues of $\nabla^2 \log p_{\mu_t}$ is equivalent to understanding the tail behavior of the localization quantity $Cov[\mathsf{r}(t)\mathbf{X}_0 | \mathbf{X}_t]$, which motivates the following section.

## 4.2 Multi-Scale Complexity: Non-Log-Concavity and Survival Function

We define a sequence of functions at various signal-to-noise ratio (SNR) scales. In a nutshell, the different scales of SNR probe the tail behavior of the localization quantity, $Cov[\mathsf{r}(t)\mathbf{X}_0 | \mathbf{X}_t]$. In turn, the localization quantity determines the landscape of the curvature of the measure $\mu_t$, a smoothed version of the original measure $\mu$ at a given SNR scale $\mathsf{r}(t)$, as shown in Equations (11)-(12). This multi-scale corruption also occurred in stochastic localization; see Montanari (2023) for the connection between stochastic localization and diffusions. We give precise complexity measures of the denoising problem, cast in quantities determined by the survival function of the random variable $Cov[\mathsf{r}(t)\mathbf{X}_0 | \mathbf{X}_t]$. As a result, these complexity measures illustrate the effective contraction or expansion rate for the diffuse-then-denoise process, covering the case of non-log-concave $\mu$'s.

Two direct consequences of the newly proposed complexity measures follow: (1) It gives rise to a fine-grained analysis for the diffuse-then-denoise process at any SNR scale, for any non-log-concave measures, to be shown in Section 4.3; (2) It motivates a simulation-based numerical tool to visualize the bottleneck SNR scale for the denoising problem, as we shall demonstrate in Section 5. Several curious phenomena are unveiled and rationalized by the new multi-scale complexity.

**Definition 4** (SNR, Localization and Survival Function). *Given an initial measure $\mu \in \mathcal{P}_2(\mathbb{R}^d)$, we define, for any SNR $\mathsf{r} \in \mathbb{R}_{\geq 0}$*

$$\mathbf{Y}_\mathsf{r} := \mathsf{r}\mathbf{X} + \mathbf{Z}, \ (\mathbf{X}, \mathbf{Z}) \sim \mu \otimes \mathcal{N}(0, I_d)$$

*where $\mathbf{X} \sim \mu$ and $\mathbf{Z} \sim \mathcal{N}(0, I_d)$ are independent.*

*Define the localization function and the associated random variable*

$$L_{\mathsf{r}}(y) = \| Cov[\mathsf{r}\mathbf{X}|\mathbf{Y}_{\mathsf{r}} = y] \|_{op} , \text{ and } \mathbf{L}_{\mathsf{r}} = \| Cov[\mathsf{r}\mathbf{X}|\mathbf{Y}_{\mathsf{r}}] \|_{op} ,$$

*and denote the survival function of $\mathbf{L}_{\mathsf{r}}$ as $s_{\mathsf{r}}(\cdot) : \mathbb{R}_+ \to [0, 1]$*

$$s_{\mathsf{r}}(u) := \mathbb{P}(\mathbf{L}_{\mathsf{r}} > u) .$$

A few remarks follow from this definition: (i) When the measure $\mu = \mathcal{L}(\mathbf{X})$ is log-concave, then by Prékopa-Leindler Inequality, for all $\mathsf{r}$, the measure $\mathcal{L}(\mathbf{Y}_{\mathsf{r}})$ is log-concave as well. Proposition 3 implies that the random variable $\mathbf{L}_{\mathsf{r}} \leq 1$. Therefore, the survival function $s_{\mathsf{r}}(1) = 0$. (ii) When the measure $\mu = \mathcal{L}(\mathbf{X})$ is non-log-concave, then the measure $\mathcal{L}(\mathbf{Y}_{\mathsf{r}})$ will be non-log-concave for some $\mathsf{r}$. For these $\mathbf{r}$'s, we know that $\exists y, \ L_{\mathsf{r}}(y) > 1$ and in turn implies $s_{\mathsf{r}}(1) = \mathbb{P}(\mathbf{L}_{\mathsf{r}} > 1) > 0$.

This property is crucial and distinguishes the difficult settings for the backward transport. The validity and effectiveness of reverting the diffusion model depends on the integrated tail of the survival function $s_{\mathsf{r}}(u), u \in [0, 1)$, which we define now.

**Definition 5** (Multi-Scale Complexity). *For $\delta \in (0, 1]$, define*

$$h_\mu(\delta, \mathsf{r}) := \int_{1-\delta}^{\infty} s_{\mathsf{r}}(u)\mathrm{d}u , \quad m_\mu(\delta, \mathsf{r}) := \frac{h_\mu(\delta, \mathsf{r})}{\delta} .$$

*Define*

$$\delta^*(\mathsf{r}) = \max \left\{ \zeta \in [0, 1] : \zeta \in \arg\min_{\delta \in [0,1]} \ m_\mu(\delta, \mathsf{r}) \right\} ,$$

*and the minimal value*

$$m^*(\mathsf{r}) := \min_{\delta \in [0,1]} \ m_\mu(\delta, \mathsf{r}) .$$

A few observations follow for the $m_\mu(\cdot, \mathsf{r}) : \delta \mapsto h_\mu(\delta, \mathsf{r})/\delta$ function, which ensures $\delta^\star(\mathsf{r})$ and $m^\star(\mathsf{r})$ are well-defined.

**Proposition 5.** *The following properties for $m_\mu(\cdot, \mathsf{r})$ hold:*

1. *Log-concave case: If $s_{\mathsf{r}}(1) = 0$, then $m_\mu(\cdot, \mathsf{r})$ is a non-decreasing function in $\delta \in [0, 1]$;*

2. *Non-log-concave case: If $s_{\mathsf{r}}(1) > 0$, then $m_\mu(\cdot, \mathsf{r})$ is either (i) non-increasing in $\delta \in [0, 1]$, or (ii) U-shaped in $\delta \in [0, 1]$, namely first non-increasing then non-decreasing.*

At a high level, if for some $\delta \in [0, 1]$ the integrated tail $\int_{1-\delta}^{\infty} s_{\mathsf{r}}(u)\mathrm{d}u$ is small, then the effective contraction of the diffuse-then-denoise process will be governed by this $\delta$. By Proposition 5, $m^*(\mathsf{r}) \geq 0$ is well-defined and, as we shall see in the next section, this will quantify the type of perturbation that the diffuse-then-denoise step can tolerate and still effectively contract.

### 4.3 Backward Expansion: Refined Analysis

In this section, we give a proof of why the multi-scale survival function at each SNR schedule $\mathsf{r}(t) = \frac{e^{-t}}{\sqrt{\beta^{-1}(1-e^{-2t})}}$ for $t \in [0, \infty]$ controls the backward expansion of the diffusion-then-denoise model. The result generalizes Theorem 2 to the case of arbitrary non-log-concave measures.

**Definition 6.** *For a measure $\mu \in \mathcal{P}_2^r(X)$, define a class of measures in reference to $\mu$,*

$$\mathcal{M}(\mu, M) := \left\{ \nu \in \mathcal{P}_2^r(X) \ : \ \frac{\sup_{x \in \mathrm{Dom}(\mu)} \|(\mathbf{t}_\mu^\nu - \mathbf{i})(x)\|^2}{\int \|(\mathbf{t}_\mu^\nu - \mathbf{i})(x)\|^2 \mathrm{d}\mu} \leq M \right\} .$$

If $M = 1$, the set $\mathcal{M}(\mu, M)$ consists of simple measures that are a location shift of $\mu$. If $M = \infty$, the set $\mathcal{M}(\mu, M) = \mathcal{P}_2^r(X)$ all Wasserstein space. Here $M$ controls the richness of perturbations around $\mu$.

**Theorem 4** (Backward Expansion: Beyond Log-Concavity). *Recall the $h_\mu(\delta, r)$ function in Definition 5. Assume $\nabla^2 \log p_{\mu_t}(x)$ has bounded eigenvalues over $x \in X$. Consider the backward OT map for the OU process as in Definition 3, then for any $\delta \in [0, 1]$*

$$\limsup_{\eta \to 0} \limsup_{\nu \in \mathcal{M}(\mu_t, M): \nu \overset{W_2}{\to} \mu_t} \frac{1}{\eta} \frac{W_2^2(\mathbf{b}_\#^{\mu, \eta} \mu_t, \mathbf{b}_\#^{\mu, \eta} \nu) - W_2^2(\mu_t, \nu)}{W_2^2(\mu_t, \nu)} \le 2 - \frac{2}{1 - e^{-2t}} \left[ \delta - M \cdot h_\mu(\delta, r(t)) \right].$$

*Here the SNR schedule $r(t)$ is defined in (12).*

To build intuition for this result, we will isolate the following term,

$$\zeta_M^*(t) = \sup_{\delta \in [0,1]} \frac{1}{1 - e^{-2t}} \left[ \delta - M \cdot h_\mu(\delta, r(t)) \right].$$

We shall extensively explore how the multi-scale complexity affects this effective curvature complexity $\zeta_M^*(t)$ with several non-log-concave examples in the next section. The curious reader may wonder whether this multi-scale complexity recovers the result obtained in the previous section for the log-concave case. The answer is yes.

In the log-concave case, namely, there exists $\zeta \ge 0$, $\nabla^2 \log p_{\mu_t}(x) \preceq -\zeta \cdot I_d$. For $\tilde{\zeta}(t) = (1 - e^{-2t}) \zeta$ and $r = r(t)$, we have $\nabla^2 \log p_{\mathbf{Y}_r}(y) \preceq -\tilde{\zeta} \cdot I_d$. In this case, $L_r(y) \le 1 - \tilde{\zeta}$ and $s_r(1 - \tilde{\zeta}) = 0$. Then, $m_\mu(\delta, r) = 0$ for all $\delta \in [0, \tilde{\zeta}]$, and

$$m^*(r) = 0, \quad \delta^*(r) = \tilde{\zeta},$$

$$\zeta_\infty^*(t) = \lim_{M \to \infty} \frac{1}{1 - e^{-2t}} \left[ \delta^*(r) - M \cdot h_\mu(\delta^*(r), r) \right] = \frac{\tilde{\zeta}}{1 - e^{-2t}} = \zeta.$$

In the non-log-concave case, for any $M < \frac{1}{m^*(r)}$, it is possible to obtain $\zeta_M^*(t) > 0$, and thus, a net contraction for the diffuse-then-denoise process. We will explore this extensively in the next section and delineate the behavior of $\zeta_M^*(t)$ for a host of fundamental non-log-concave distributions.

## 5 Examples

Theorem 4 depends on an integrated survival function $h_\mu(\delta, r)$ whose shape is difficult to guess. Luckily, its complete behavior can be captured by understanding the following quantities.

$$s_r(u) = \mathbb{P}(\mathbf{L}_r > u), \quad m^*(r) = \min_{\delta \in [0,1]} m_\mu(\delta, r), \quad \delta^*(r) = \max \left\{ \zeta \in [0, 1] : \zeta \in \underset{\delta \in [0,1]}{\arg\min} \ m_\mu(\delta, r) \right\}.$$

We will methodically calculate and plot these objects for a range of fundamental distributions. Given a target distribution, empirical or with explicit form, we can use a Monte Carlo simulation method to visualize the functions above.

- For any empirical measure $\mu_0$, (15) defines an expression for the empirical version of $\mathbf{L}_r(y)$.

- One can simulate $\mathbf{Y}_r$ by sampling $rX + \mathcal{N}(0, 1)$, for $X \sim \mu_0$.

- Then one can obtain the empirical survival function, $s_r(u)$. $m^*(r)$, $\delta^*(r)$ and $\zeta_M^*(t)$ follow.

Capturing this behavior at precise time scales will allow us to ascertain (1) when contraction is easy, and if not, (2) at what SNR, $r$, the process transitions to a non-log-concave setting. In the following, we will restrict ourselves to the OU process, with $\beta = 1$, and in one dimension.

### 5.1 Warm-Up: Log-Concave

Suppose the target measure, $\mu_0$ is log-concave. Via Prékopa-Leindler Inequality, the full chain of measures generated by the OU process admits log-concavity. That is, there exists a sequence of non-negative constants $\zeta(r) > 0$ such that

$$\nabla^2 \log p_{\mathbf{Y}_r}(y) \preceq -\zeta(r), \quad \forall r \geq 0.$$

We know from the previous section that this implies,

$$m^*(r) = 0, \quad \delta^*(r) = \zeta(r), \quad \zeta_\infty^*(t) = \frac{\delta^*(r(t))}{1 - e^{-2t}}.$$

We visualize and validate this result for three log-concave distributions.

*Example* 1 (Point Mass). Consider the simplest case where the target measure $\mu_0$ is a Dirac measure, $\delta_0$. We assess the localization of the backward transport by investigating the random covariance, $\mathbf{L}_r$.

$$-\nabla^2 \log p_{\mathbf{Y}_r}(y) = 1, \quad \implies \mathbf{L}_r \equiv 0.$$

In this simple case, $\mathbf{L}_r$ is a point mass at 0. The backward transport localizes completely. It follows that $s_r(u) = 0$. Therefore,

$$m^*(r) = 0, \quad \delta^*(r) = 1, \quad \zeta_\infty^*(t) = \frac{1}{1 - e^{-2t}}.$$

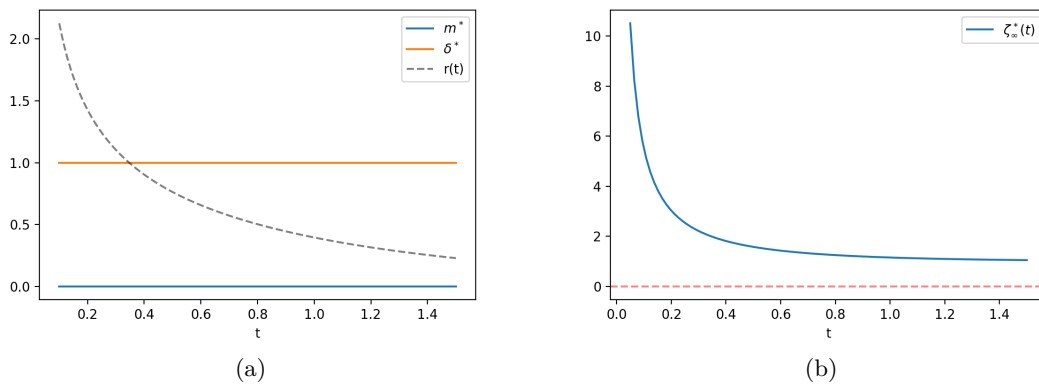

(a)  (b)

Figure 3: (a) $m^*(r), \delta^*(r)$. (b) $\zeta^*(t)$.

*Example* 2 (Normal). Consider now that $\mu_0$ is a normal distribution $\mathcal{N}(m, \sigma^2)$. Again, we calculate the localization quantity,

$$-\nabla^2 \log p_{\mathbf{Y}_r}(y) = \frac{1}{\sigma^2 r^2 + 1}, \quad \implies \mathbf{L}_r = \frac{\sigma^2 r^2}{\sigma^2 r^2 + 1}.$$

$\mathbf{L}_r$ is a point mass with location depending on the SNR, $r$. In turn, the survival function is not 0, but a step function with an explicit threshold,

$$s_r(u) = \begin{cases} 1, & \forall u \in [0, \frac{\sigma^2 r^2}{\sigma^2 r^2 + 1}) \\ 0, & \forall u \in [\frac{\sigma^2 r^2}{\sigma^2 r^2 + 1}, \infty) \end{cases}$$

We plot this in Figure 4 (a). The backward transport localizes completely as $r \to 0$ (or equivalently $t \to \infty$).

For any $r$, we can find $\delta \in [0, 1]$ such that $m_\mu(\delta, r) = 0$. It follows,

$$m^*(r) = 0, \quad \delta^*(r) = \frac{1}{\sigma^2 r^2 + 1}, \quad \zeta_\infty^*(t) = \frac{1}{1 + e^{-2t}(\sigma^2 - 1)}.$$

We confirm a variety of $\sigma^2$ choices in Figure 4 (b) and (c).

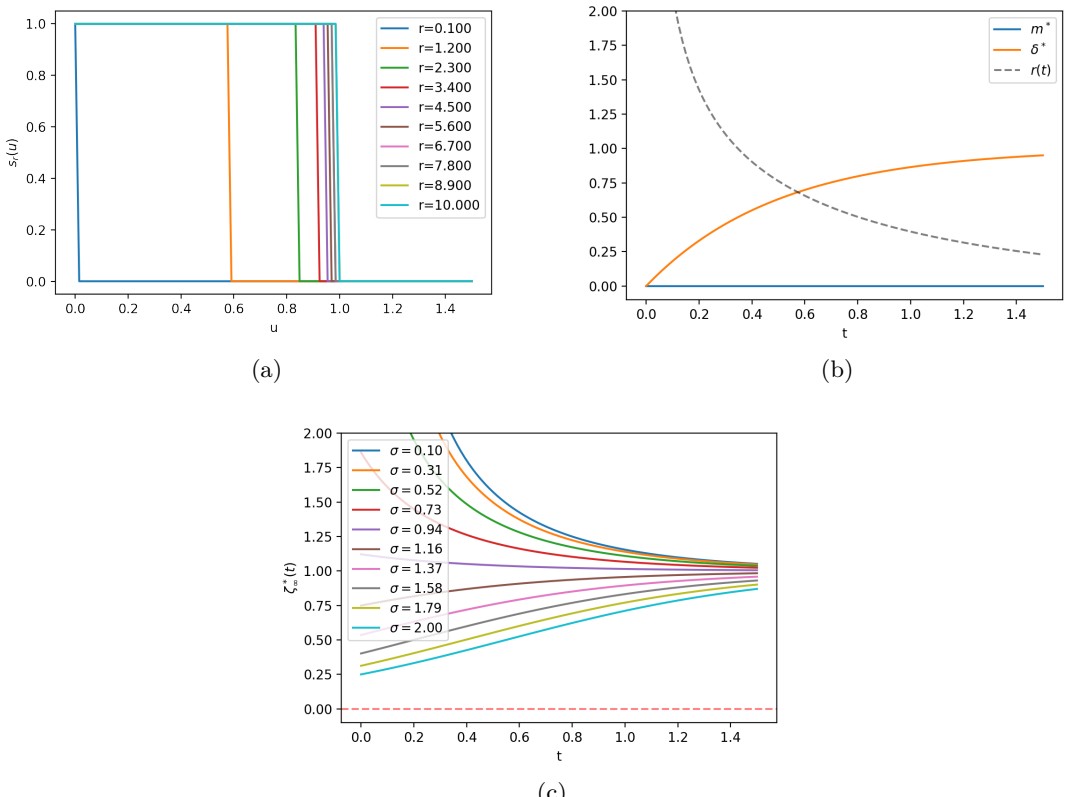

$$(a) \qquad\qquad\qquad\qquad\qquad (b)$$

$$(c)$$

Figure 4: (a) $s_{\mathsf{r}}(u)$. (b) $m^*(\mathsf{r}), \delta^*(\mathsf{r})$. (c) $\zeta^*(t)$.

*Example* 3 (Uniform). Consider the case when the target distribution is uniform $\mu_0 = \mathrm{Unif}(-1, 1)$. As before, we gauge the localization via the random covariance of the backward transport, $\mathbf{L}_{\mathsf{r}}$.

$$\nabla^2 \log p_{\mathbf{Y}_{\mathsf{r}}}(y) = \left[ \frac{-(y+\mathsf{r})\phi(y+\mathsf{r}) + (y-\mathsf{r})\phi(y-\mathsf{r})}{\Phi(y+\mathsf{r}) - \Phi(y-\mathsf{r})} - \left( \frac{\phi(y+\mathsf{r}) - \phi(y-\mathsf{r})}{\Phi(y+\mathsf{r}) - \Phi(y-\mathsf{r})} \right)^2 \right],$$
$$L_{\mathsf{r}}(Y_{\mathsf{r}}) = 1 + \nabla^2 \log p_{Y_{\mathsf{r}}}(Y_{\mathsf{r}}) .$$

Based on this characterization we simulate $s_{\mathsf{r}}(u)$, $m^*(\mathsf{r})$, $\delta^*(\mathsf{r})$, $\zeta^*_\infty(t)$. As expected we see,

$$s_{\mathsf{r}}(1) = 0, \quad m^*(\mathsf{r}) = 0, \quad \zeta^*_\infty(t) > 0.$$

We recover the same qualitative behavior as in the preceding examples (point mass, Gaussian). However, the fine-grained behavior is remarkably different; see Figure 5 (c). Note that though $\zeta^*_\infty(t) > 0$, we have the non-monotonic behavior of the effective contract at different time scales. The slowest effective contraction is happening in some small time scales.

The uniting technicality that allows this is:

$$\mathbf{L}_{\mathsf{r}} \le 1, \iff s_{\mathsf{r}}(1) = 0, \quad \forall \mathsf{r} \ge 0.$$

By Proposition 3, this is exactly saying that the curvature, $\nabla^2 \log p_{\mathbf{Y}_{\mathsf{r}}}(y) \le 0$.

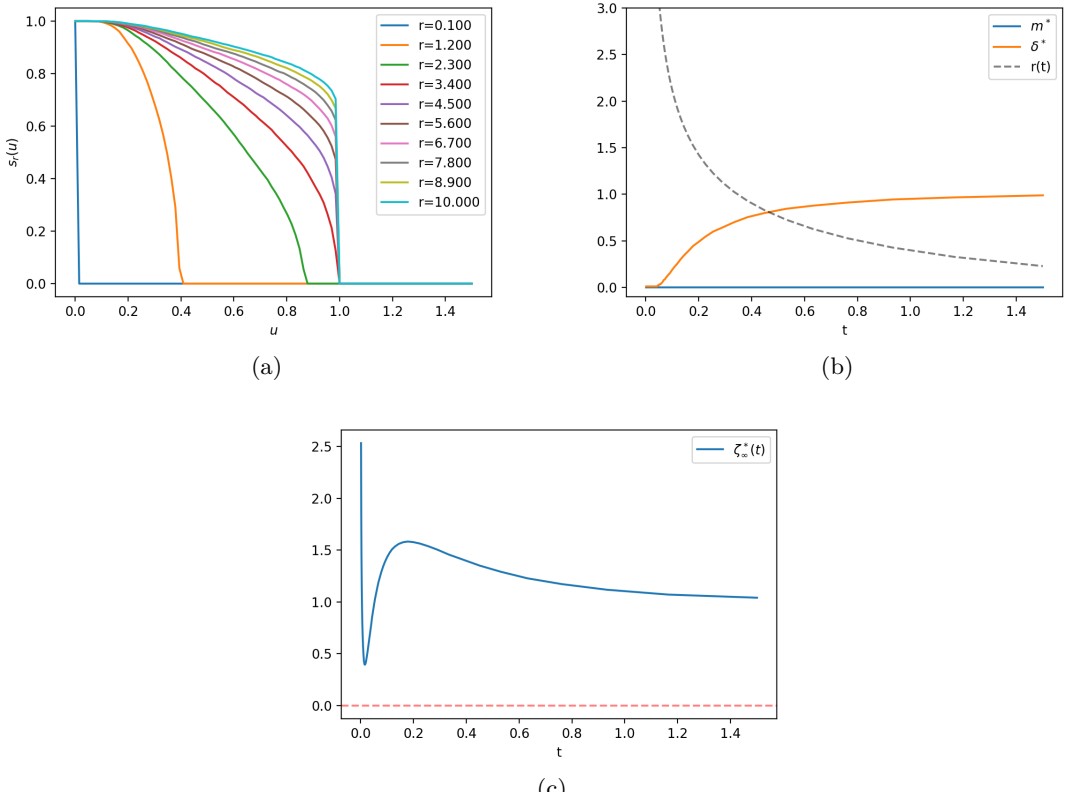

Figure 5: (a) $s_{\mathsf{r}}(u)$. (b) $m^*(\mathsf{r}), \delta^*(\mathsf{r})$. (c) $\zeta^*(t)$.

## 5.2 Beyond Log-Concavity

With general non-log-concave $\mu$ as initialization, the Ornstein-Uhlenbeck process at time $t$ may not have log-concavity. Indeed, $\nabla^2 \log p_{\mathbf{Y}_{\mathsf{r}}}(y)$ is not uniformly upper bounded by 0, or equivalently, the random covariance $\mathbf{L}_{\mathsf{r}}$ cannot be uniformly bounded from above by 1. We separate from the regime of the previous three examples, all of which exploited this property to show contraction at all scales $\mathsf{r}$.

It is important to understand (i) when the backward chain enters this non-log-concave regime, and (ii) the expansion behavior in this regime. To that end, we make use of the following notation:

$$\mathsf{r}^* = \max\{\mathsf{r} : s_{\mathsf{r}}(1) = 0\}.$$

*Example* 4 (Two Point Mass). We consider $X_0 \sim \frac{1}{2}\delta_{-\mu} + \frac{1}{2}\delta_\mu$ for some $\mu > 0$. We assess the localization via the random variance, $\mathbf{L}_{\mathsf{r}}$.

$$L_{\mathsf{r}}(y) = (\mathsf{r}\mu)^2 \left[1 - \left(\frac{\phi(y - \mathsf{r}\mu) - \phi(y + \mathsf{r}\mu)}{\phi(y - \mathsf{r}\mu) + \phi(y + \mathsf{r}\mu)}\right)^2\right], \quad \sup_y L_{\mathsf{r}}(y) = L_{\mathsf{r}}(0) = (\mathsf{r}\mu)^2.$$

We see immediately, $\mathsf{r}^* = \mu^{-1}$. We can explicitly calculate $s_{\mathsf{r}}(u)$.

$$s_{\mathsf{r}}(u) = \Phi\left(\frac{1}{2(\mathsf{r}\mu)} \log \frac{\mathsf{r}\mu + \sqrt{(\mathsf{r}\mu)^2 - u}}{\mathsf{r}\mu - \sqrt{(\mathsf{r}\mu)^2 - u}} + \mathsf{r}\mu\right) + \Phi\left(\frac{1}{2(\mathsf{r}\mu)} \log \frac{\mathsf{r}\mu + \sqrt{(\mathsf{r}\mu)^2 - u}}{\mathsf{r}\mu - \sqrt{(\mathsf{r}\mu)^2 - u}} - \mathsf{r}\mu\right) - 1 \quad (13)$$

$$s_{\mathsf{r}}(u) = 0, \quad u \in [(\mathsf{r}\mu)^2, \infty). \quad (14)$$

(i) $(\mathsf{r} \leq \mu^{-1})$: In this setting, we expect log-concave behavior. By (14),

$$m^*(\mathsf{r}) = 0, \quad \delta^*(\mathsf{r}) = 1 - (\mathsf{r}\mu)^2, \quad \zeta_\infty^*(t) = \frac{1 - (\mathsf{r}(t)\mu)^2}{1 - e^{-2t}} \overset{t\to\infty}{\Rightarrow} 1.$$

(ii) $(\mathsf{r} \to \infty)$: For extremely high SNR, we can see $s_\mathsf{r}(u) \to 0$. This phenomenon is qualitatively the same as log-concavity and can be rationalized as essentially recovering the single-point mass behavior in the limit. In particular,

$$m^*(\mathsf{r}) \to 0, \quad \delta^*(\mathsf{r}) \to 1, \quad \zeta_\infty^*(t) \to \infty.$$

(iii) Contraction should be hardest at mid-range SNR, where we no longer have log-concavity. The behavior is simulated and displayed in Figure 6 (c). In this setting, we can tolerate a complexity $M \leq \frac{1}{m^*(\mathsf{r})} < \infty$ in order for non-expansion $\zeta_M^*(t) \geq 0$. This is visualized in Figure 6 (d). Note the non-monotonic behavior of the curvature complexity $\zeta_M^*(t)$: there seems to be a mid-range of time at which the diffuse-then-denoise chain could expand, when the type of perturbation is complex with large $M$.

The conclusion is not pessimistic. For most of the process, regardless of $M$, $\zeta_M^*(t)$ is positive and contraction occurs. We remind the reader that $M$ is a proposed upper bound in Definition 6. In our numerical examples, this is typically small. Refer to Figure 7 to see the success of diffuse-then-denoise in this case.

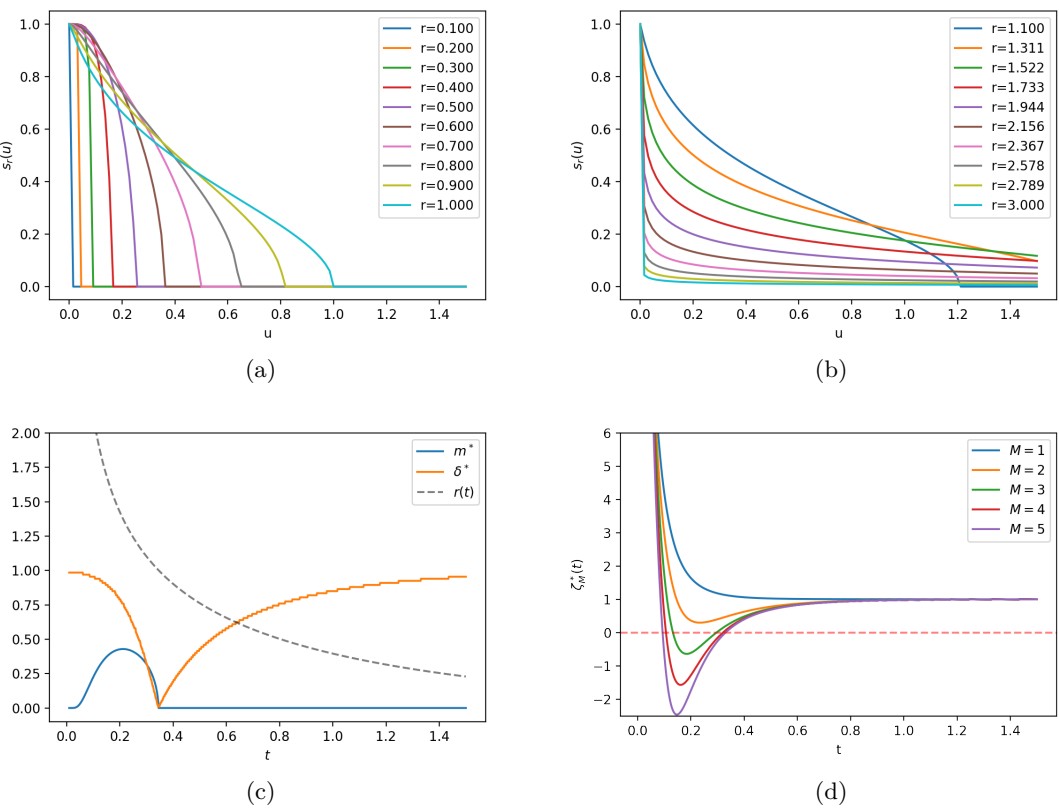

Figure 6: (a) $s_\mathsf{r}(u)$ Low SNR. (b) $s_\mathsf{r}(u)$ High SNR. (c) $m^*(\mathsf{r}), \delta^*(\mathsf{r})$. (d) $\zeta_M^*(t)$.

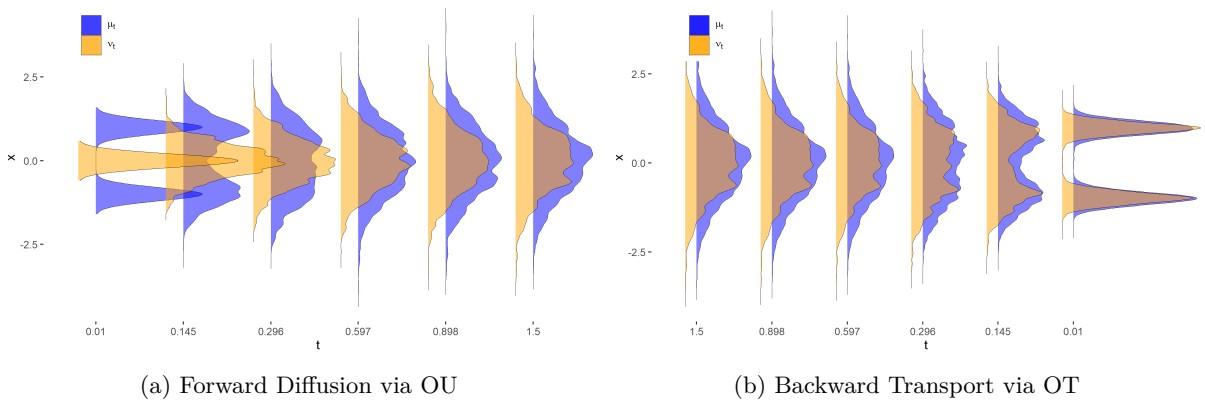

(a) Forward Diffusion via OU

(b) Backward Transport via OT

Figure 7: (a) Forward diffusion initialized at $\mu_0 = 0.5\delta_1 + 0.5\delta_{-1}$, $\nu_0 = \delta_0$. $T = 100$, $\eta = 0.01$. (b) Backward transport initialized at $\mu_T$, $\nu_T$, and applying the backward OT map $\mathbf{b}^{\mu,\eta}$.

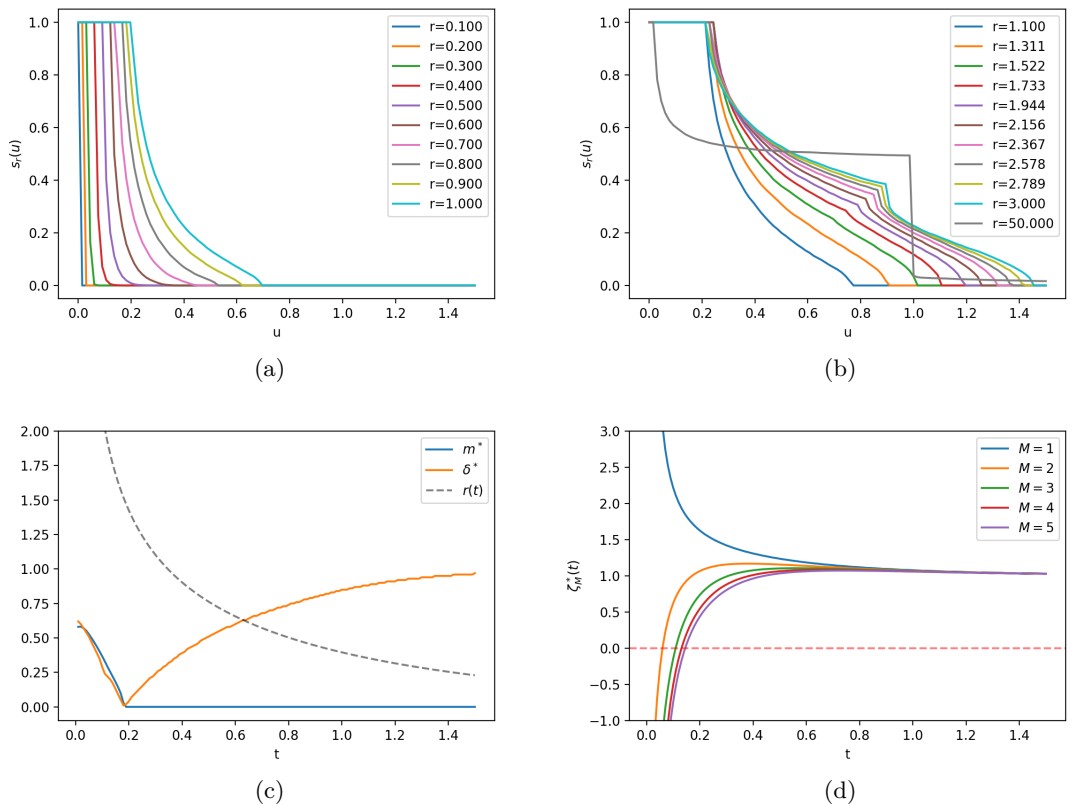

Figure 8: (a) $s_{\mathsf{r}}(u)$ Low SNR. (b) $s_{\mathsf{r}}(u)$ High SNR. (c) $m^*(\mathsf{r}), \delta^*(\mathsf{r})$. (d) $\zeta_M^*(t)$.

*Example* 5 (Mixture of Point Mass and Gaussian). We now consider diffuse-then-denoise for the target distribution; $\frac{1}{2}\delta_0 + \frac{1}{2}\mathcal{N}(0,1)$. Let $v = \sqrt{r^2 + 1}$. We first calculate the covariance,

$$L_{\mathsf{r}}(y) = 1 + \frac{v^{-3}\left(y^2/v^2 - 1\right)\phi\left(y/v\right) + (y^2 - 1)\phi(y)}{v\phi\left(y/v\right) + \phi(y)} - \left(\frac{v^{-3}y\phi\left(y/v\right) + y\phi(y)}{v\phi\left(y/v\right) + \phi(y)}\right)^2.$$

We simulate the survival function $s_{\mathsf{r}}(u) = \mathbb{P}(\mathbf{L}_{\mathsf{r}} > u)$ via Monte Carlo, see Figures 8 (a) and (b). A lack of log-concavity is clear for high SNR.

(i) ($r \leq r^*$): We expect $m^*(r) = 0$. While we don't have an explicit form for $r^*$, simulations in Figure 8 validate this. See also Monte Carlo-based simulations for $\delta^*$, $\zeta_\infty^*$.

(ii) ($r > r^*$): We move away from the log-concave regime for large SNR (small $t$). We conjecture that as $r \to \infty$, $m^*(r) \to \frac{1}{2}$. To see this, note that $s_r(t)$ approaches a step function form with a step at 0.5. In this setting, the behavior of $\zeta_M^*$ is captured in Figure 8 (d). We note that regardless of $M$, the range of time when the process effectively expands is minimal compared to the full process.

Refer to Figure 9 to see the success of diffuse-then-denoise in this case.

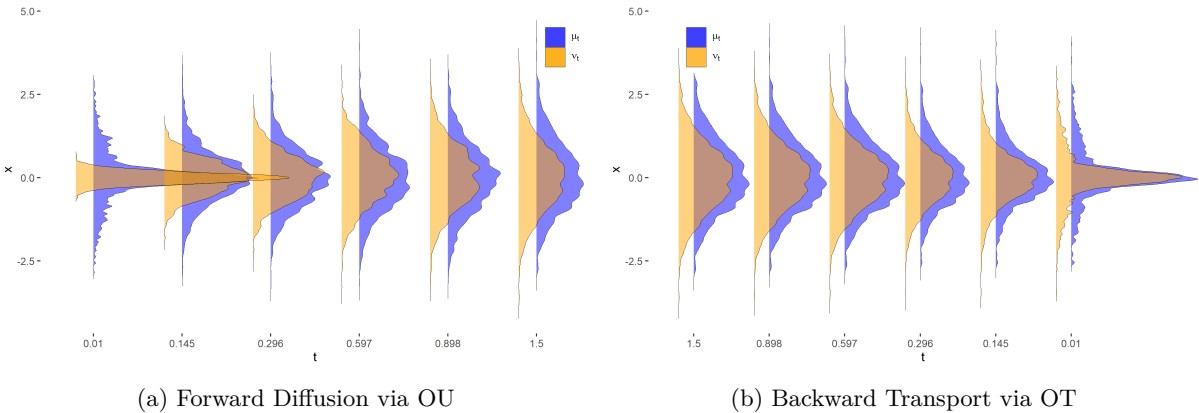

(a) Forward Diffusion via OU  (b) Backward Transport via OT

Figure 9: (a) Forward diffusion initialized at $\mu_0 = 0.5\delta_0 + 0.5\mathcal{N}(0,1)$, $\nu_0 = \delta_0$. $T = 100$, $\eta = 0.01$. (b) Backward transport initialized at $\mu_T$, $\nu_T$, and applying the backward OT map $\mathbf{b}^{\mu,\eta}$.

*Example* 6 (Mixture of Gaussian, Heterogeneous Variance). Consider the general Gaussian Mixture given by,

$$X_0 \sim \sum_{i=1}^m p_i \cdot \mathcal{N}(\mu_i, \sigma_i^2)$$

In Figure 10, we consider a specific formulation: $0.1\mathcal{N}(-4,1) + 0.2\mathcal{N}(-2,0.5) + 0.4\mathcal{N}(2,0.5) + 0.3\mathcal{N}(4,1)$. Defining $v_i = \sqrt{r^2\sigma_i^2 + 1}$, $t_i = v_i^{-1}(y - r\mu_i)$, for $i \in [m]$, we derive the conditional covariance:

$$L_r(y) = 1 + \frac{\sum_i p_i v_i^{-3}(t_i^2 - 1)\phi(t_i)}{\sum_i p_i v_i \phi(t_i)} - \left( \frac{-\sum_i p_i v_i^{-2} t_i \phi(t_i)}{\sum_i p_i v_i \phi(t_i)} \right)^2 \tag{15}$$

The survival function $s_r(u) = \mathbb{P}(\mathbf{L}_r > u)$ is simulated via Monte Carlo and is displayed in Figures 10 (a) and (b). As soon as the SNR is not small, we are no longer in the log-concave setting.

(i) ($r \leq r^*$): We expect $m^*(r) = 0$. Simulations in Figure 10 show a flat $m^*$ for sufficiently large $r$. Even for mid-range SNR, we diverge from this log-concave setting.

(ii) ($r > r^*$): The shape of $\zeta_M^*$ is captured in Figure 10 (d). In contrast to the previous two cases, this region is not so narrow. For a chosen noise schedule, diffuse-then-denoise still seems successful for sampling in this setting, see Figure 11.

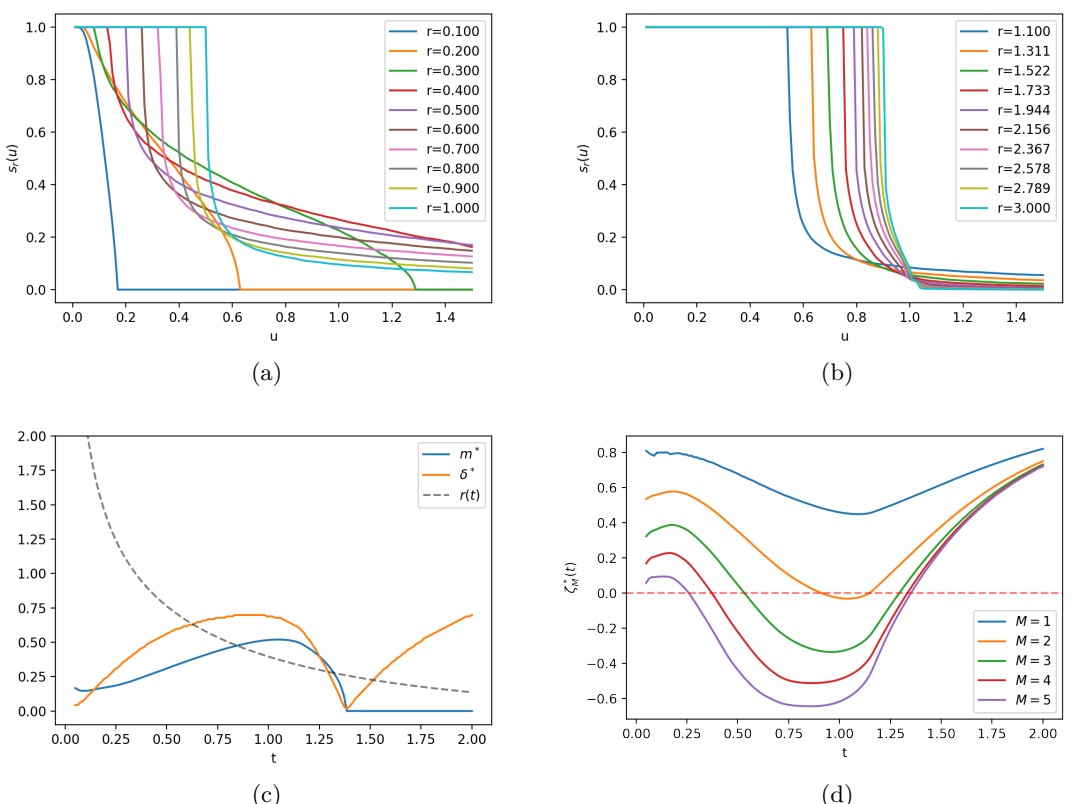

Figure 10: (a) $s_{\mathsf{r}}(u)$ Low SNR. (b) $s_{\mathsf{r}}(u)$ High SNR. (c) $m^*(\mathsf{r}), \delta^*(\mathsf{r})$. (d) $\zeta_M^*(t)$.

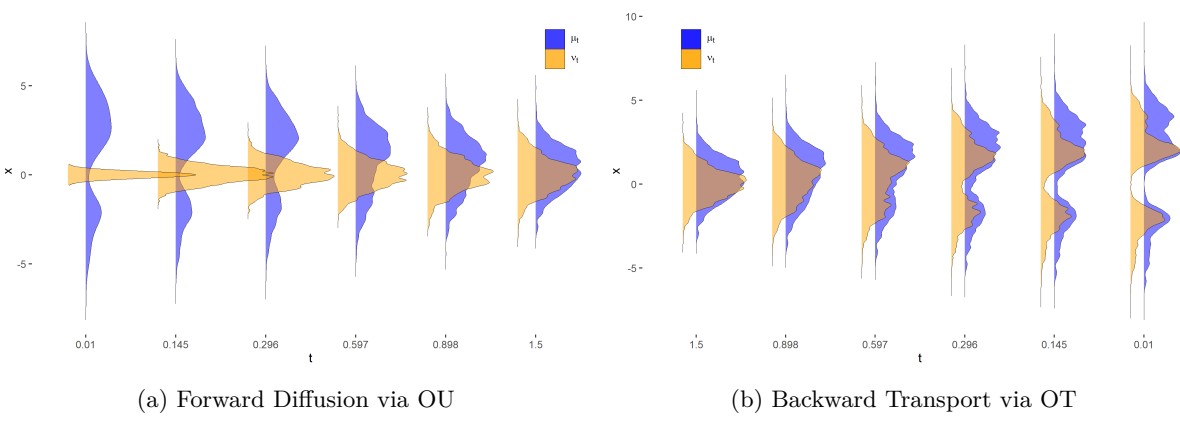

(a) Forward Diffusion via OU    (b) Backward Transport via OT

Figure 11: (a) Forward diffusion initialized at $\mu_0 = 0.1\mathcal{N}(-4,1) + 0.2\mathcal{N}(-2,0.5) + 0.4\mathcal{N}(2,0.5) + 0.3\mathcal{N}(4,1)$, $\nu_0 = \delta_0$. $T = 100$, $\eta = 0.01$. (b) Backward transport initialized at $\mu_T$, $\nu_T$, and applying the backward OT map $\mathbf{b}^{\mu,\eta}$.

## Acknowledgements

TL acknowledges the support from the NSF Career Award (DMS-2042473) and the Wallman Society of Fellows at the University of Chicago.

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

# A   Proofs

The Appendix is organized as follows.

Section A.1 gathers the technical preparations for denoising and localization, where we prove the results in Section 2. We derive the optimal transport denoising map and explain how the curvature function appears from a localization perspective.

Section A.2 presents analyses of the forward and backward processes, where we prove the results in Section 3. We demonstrate how the curvature function influences backward denoising via a sensitivity analysis under the Wasserstein metric. We also study the diffuse-then-denoise process, showing that adding noise then denoising can be beneficial in the presence of negative curvature.

Section A.3 provides proofs for the results in Section 4. We extend beyond the log-concave setting by defining and analyzing a multi-scale complexity that captures an average notion of curvature across all time scales. This multi-scale complexity plays a key role in the sensitivity analysis of the diffusion model under the Wasserstein metric.

## A.1   Proofs for Section 2

*Proof of Proposition 1.* By Lemma 10.1.2 in Ambrosio et al. (2008),

$$\frac{1}{\eta}(\mathbf{t}^{\mu_t}_{\mu_{t+\eta}} - \mathbf{i}) \in \partial \mathcal{G}(\mu_{t+\eta})$$

where $\partial \mathcal{G}(\mu_{t+\eta})$ is the strong subdifferential. Recall that $\mathcal{G}(\nu) = \mathcal{F}(\nu) + \beta^{-1}\mathcal{E}(\nu)$. For a given $\nu$ with density $\nu = \rho \cdot \mathcal{L}^d$, by Lemma 10.4.1 in Ambrosio et al. (2008), we know any $\boldsymbol{\xi}(x) \in L^2(\nu; \mathbb{R}^d)$ in $\partial \mathcal{G}(\nu)$ admits the representation

$$\boldsymbol{\xi}(x) = \nabla \frac{\delta \mathcal{G}}{\delta \rho} = \nabla f(x) + \beta^{-1} \nabla \log \rho(x), \text{ for } \nu\text{-a.e. } x \in \mathbb{R}^d .$$

Take $\nu = \mu_{t+\eta}$, we finish the proof. □

*Proof of Proposition 2.* This result follows from Tweedie's formula (Robbins, 1992). The conditional distribution of $\mathbf{Y} = y$ given $\mathbf{X}$ admits the following density: $p_{\mathbf{Y}}(y|\mathbf{X}) := \frac{\mathrm{d}}{\mathrm{d}y}P(\mathbf{Y} \le y|\mathbf{X}) = \frac{1}{\sigma}\phi((y - \mathbf{X})/\sigma)$ where $\phi : \mathbb{R}^d \to \mathbb{R}$ is the density of $d$-dimensional standard normal. This implies

$$\nabla p_{\mathbf{Y}}(y|\mathbf{X}) = -\frac{1}{\sigma^2}(y - \mathbf{X})p_{\mathbf{Y}}(y|\mathbf{X}) \quad \text{and} \quad \nabla p_{\mathbf{Y}}(y) = -\frac{1}{\sigma^2}\mathbb{E}[(y - \mathbf{X})p_{\mathbf{Y}}(y|\mathbf{X})]. \tag{16}$$

Therefore, we obtain

$$\frac{\nabla p_{\mathbf{Y}}(y)}{p_{\mathbf{Y}}(y)} = -\frac{1}{\sigma^2}\mathbb{E}[(y-\mathbf{X})\frac{p_{\mathbf{Y}}(y|\mathbf{X})}{p_{\mathbf{Y}}(y)}] = -\frac{1}{\sigma^2}\{y - \mathbb{E}[\mathbf{X}|\mathbf{Y} = y]\}\,,$$

where we used the fact that $\mathbb{E}[h(\mathbf{X})\frac{p_{\mathbf{Y}}(y|\mathbf{X})}{p_{\mathbf{Y}}(y)}] = \mathbb{E}[h(\mathbf{X})|\mathbf{Y} = y]$ for any integrable function $h$ with respect to $\mathbf{X}$. This completes the proof. $\qquad\square$

*Proof of Proposition 3.* Taking the derivative of (16) with respect to $y$, we have

$$\nabla^2 p_{\mathbf{Y}}(y|\mathbf{X}) = \Big[-\frac{1}{\sigma^2}I_d + \frac{1}{\sigma^4}(y-\mathbf{X})(y-\mathbf{X})^\top\Big]p_{\mathbf{Y}}(y|\mathbf{X})\,,$$

and $\nabla^2 p_{\mathbf{Y}}(y) = \mathbb{E}\Big[\big(-\frac{1}{\sigma^2}I_d + \frac{1}{\sigma^4}(y-\mathbf{X})(y-\mathbf{X})^\top\big)p_{\mathbf{Y}}(y|\mathbf{X})\Big]$. This gives

$$\begin{aligned}
\frac{\nabla^2 p_{\mathbf{Y}}(y)}{p_{\mathbf{Y}}(y)} &= \mathbb{E}\Big[\big(-\frac{1}{\sigma^2}I_d + \frac{1}{\sigma^4}(y-\mathbf{X})(y-\mathbf{X})^\top\big)\frac{p_{\mathbf{Y}}(y|\mathbf{X})}{p_{\mathbf{Y}}(y)}\Big]\,,\\
&= -\frac{1}{\sigma^2}I_d + \frac{1}{\sigma^4}\mathbb{E}[(y-\mathbf{X})(y-\mathbf{X})^\top|\mathbf{Y} = y]\,.
\end{aligned}$$

Combined with the result $\frac{\nabla p_{\mathbf{Y}}(y)}{p_{\mathbf{Y}}(y)} = -\frac{1}{\sigma^2}(\mathbb{E}[y-\mathbf{X}|\mathbf{Y} = y])$ from Proposition 2, we obtain

$$\begin{aligned}
\nabla^2 \log p_{\mathbf{Y}}(y) &= \frac{\nabla^2 p_{\mathbf{Y}}(y)}{p_{\mathbf{Y}}(y)} - \frac{\nabla p_{\mathbf{Y}}(y)}{p_{\mathbf{Y}}(y)}\Big(\frac{\nabla p_{\mathbf{Y}}(y)}{p_{\mathbf{Y}}(y)}\Big)^\top\\
&= -\frac{1}{\sigma^2}I_d + \frac{1}{\sigma^4}\mathbb{E}[(y-\mathbf{X})(y-\mathbf{X})^\top|\mathbf{Y} = y]\\
&\quad -\frac{1}{\sigma^4}(\mathbb{E}[y-\mathbf{X}|\mathbf{Y} = y])(\mathbb{E}[y-\mathbf{X}|\mathbf{Y} = y])^\top\\
&= -\frac{1}{\sigma^2}I_d + \frac{1}{\sigma^4}Cov[y-\mathbf{X}|\mathbf{Y} = y]\,.
\end{aligned}$$

With $Cov[y-\mathbf{X}|\mathbf{Y} = y] = Cov[\mathbf{X}|\mathbf{Y} = y] = \sigma^2\,Cov[\sigma^{-1}\mathbf{X}|\mathbf{Y}]$, we complete the proof. $\qquad\square$

*Proof of Proposition 4.* Using the integration by parts, we have

$$\begin{aligned}
\int tr[\nabla^2 \log p_\nu(x)]\mathrm{d}\nu(x) &= \sum_{i=1}^d \int p_\nu(x)\frac{\partial^2}{\partial x_i}\log p_\nu(x)\mathrm{d}x\,,\\
&= -\sum_{i=1}^d \int \frac{\partial}{\partial x_i}p_\nu(x)\frac{\partial}{\partial x_i}\log p_\nu(x)\mathrm{d}x\,,\\
&= -\int \|\nabla \log p_\nu(x)\|^2 p_\nu(x)\mathrm{d}x\,.
\end{aligned}$$

$\qquad\square$

## A.2  Proofs for Section 3

*Proof of Theorem 1.* Take any two $\mu,\nu \in \mathcal{P}_2^r(X)$, we have by convexity of $\mathcal{E}(\cdot)$

$$\begin{aligned}
\mathcal{E}(\mu) - \mathcal{E}(\nu) &\geq \int \langle \nabla \log p_\nu, \mathbf{t}_\nu^\mu - \mathbf{i}\rangle \mathrm{d}\nu = \langle \nabla \log p_\nu(\mathbf{t}_\mu^\nu), \mathbf{i} - \mathbf{t}_\mu^\nu\rangle \mathrm{d}\mu\,,\\
\mathcal{E}(\nu) - \mathcal{E}(\mu) &\geq \int \langle \nabla \log p_\mu, \mathbf{t}_\mu^\nu - \mathbf{i}\rangle \mathrm{d}\mu\,.
\end{aligned}$$

Therefore, adding up these two inequalities, we have

$$\int \langle \nabla \log p_\mu - \nabla \log p_\nu(\mathbf{t}^\nu_\mu), \mathbf{t}^\nu_\mu - \mathbf{i} \rangle \mathrm{d}\mu \leq 0 \ . \tag{17}$$

Now denote $\mathbf{t} := \mathbf{t}^{\nu_t}_{\mu_t}$, we know $(\mathbf{f}^{\mu,\eta}, \mathbf{f}^{\nu,\eta} \circ \mathbf{t})_{\#}\mu_t \in \Pi(\mu_t, \nu_t)$ where $\mathbf{f}^{\mu,\eta}, \mathbf{f}^{\nu,\eta}$ are defined in (8),

$$W_2^2(\mu_{t+\eta}, \nu_{t+\eta}) = \inf_{\pi \in \Pi(\mu_{t+\eta}, \nu_{t+\eta})} \int \|x - y\|^2 \mathrm{d}\pi(x, y) \ ,$$

$$\leq \int \|(\mathbf{i} - \eta \nabla f - \eta \beta^{-1} \nabla \log p_{\mu_t}) - (\mathbf{t} - \eta \nabla f(\mathbf{t}) - \eta \beta^{-1} \nabla \log p_{\nu_t}(\mathbf{t}))\|^2 \mathrm{d}\mu_t \ ,$$

$$= \int \|\mathbf{i} - \mathbf{t}\|^2 \mathrm{d}\mu_t - 2\eta \int \langle \mathbf{i} - \mathbf{t}, \nabla f - \nabla f(\mathbf{t}) \rangle \mathrm{d}\mu_t$$

$$+ 2\eta \beta^{-1} \int \langle \nabla \log p_{\mu_t} - \nabla \log p_{\nu_t}(\mathbf{t}), \mathbf{t} - \mathbf{i} \rangle \mathrm{d}\mu_t$$

$$+ O(\eta^2) \ ,$$

$$\leq (1 - 2\eta\lambda) W_2^2(\mu_t, \nu_t) + O(\eta^2) \ .$$

Here we use (17) and the fact $\langle \nabla f(x) - \nabla f(y), x - y \rangle \geq \lambda \|x - y\|^2$, for all $x, y$. Rearrange and then let $\eta \to 0$, we prove the theorem. □

*Proof of Theorem 2.* First, for any $\nu \in \mathcal{P}_2^r(X)$, define two quantities

$$\epsilon := \|\mathbf{t}^\nu_{\mu_t} - \mathbf{i}\|_{L^2(\mu_t;\mathbb{R}^d)} = W_2(\nu, \mu_t) \ ,$$

$$\boldsymbol{\xi} := (\mathbf{t}^\nu_{\mu_t} - \mathbf{i})/\|\mathbf{t}^\nu_{\mu_t} - \mathbf{i}\|_{L^2(\mu_t;\mathbb{R}^d)} \ ,$$

Then $\mathbf{t}^\nu_{\mu_t} = \mathbf{i} + \epsilon \boldsymbol{\xi}$. For any $\nu \in \mathcal{P}_2^r(X)$, we have

$$W_2^2(\mathbf{b}^{\mu,\eta}_\# \mu_t, \mathbf{b}^{\mu,\eta}_\# \nu) \leq \int \|\mathbf{b}^{\mu,\eta} \circ (\mathbf{i} + \epsilon\boldsymbol{\xi}) - \mathbf{b}^{\mu,\eta} \circ \mathbf{i}\|^2 \mathrm{d}\mu_t \ ,$$

$$= \int \|\mathbf{b}^{\mu,\eta}(x + \epsilon\boldsymbol{\xi}(x)) - \mathbf{b}^{\mu,\eta}(x)\|^2 \mathrm{d}\mu_t(x) \ .$$

Define an auxiliary function $g_{\boldsymbol{\xi}}(\epsilon) := \int \|\mathbf{b}^{\mu,\eta}(x + \epsilon\boldsymbol{\xi}(x)) - \mathbf{b}^{\mu,\eta}(x)\|^2 \mathrm{d}\mu_t(x)$, we can verify that

$$g'_{\boldsymbol{\xi}}(0) = 0, \ \lim_{\epsilon \to 0} \frac{g_{\boldsymbol{\xi}}(\epsilon)}{\epsilon} = 0 \ .$$

We also know that

$$g'_{\boldsymbol{\xi}}(\epsilon) = \int 2 \langle \nabla \mathbf{b}^{\mu,\eta}(x + \epsilon\boldsymbol{\xi}(x))\boldsymbol{\xi}(x), \mathbf{b}^{\mu,\eta}(x + \epsilon\boldsymbol{\xi}(x)) - \mathbf{b}^{\mu,\eta}(x) \rangle \mathrm{d}\mu_t(x) \ ,$$

$$\leq \frac{\int \|\mathbf{b}^{\mu,\eta}(x + \epsilon\boldsymbol{\xi}(x)) - \mathbf{b}^{\mu,\eta}(x)\|^2 \mathrm{d}\mu_t(x)}{\epsilon} + \epsilon \cdot \int \|\nabla \mathbf{b}^{\mu,\eta}(x + \epsilon\boldsymbol{\xi}(x))\boldsymbol{\xi}(x)\|^2 \mathrm{d}\mu_t(x) \ ,$$

$$\leq \frac{g_{\boldsymbol{\xi}}(\epsilon)}{\epsilon} + \epsilon \cdot \sup_x \|\nabla \mathbf{b}^{\mu,\eta}(x)\|^2_{op} \cdot \int \|\boldsymbol{\xi}(x)\|^2 \mathrm{d}\mu_t(x) \ .$$

Notice $\int \|\boldsymbol{\xi}(x)\|^2 \mathrm{d}\mu_t(x) = 1$, and divide both sides by $\epsilon$, we obtain

$$\frac{\mathrm{d}}{\mathrm{d}\epsilon} \left( \frac{g_{\boldsymbol{\xi}}(\epsilon)}{\epsilon} \right) = \frac{g'_{\boldsymbol{\xi}}(\epsilon)\epsilon - g_{\boldsymbol{\xi}}(\epsilon)}{\epsilon^2} \leq \sup_x \|\nabla \mathbf{b}^{\mu,\eta}(x)\|^2_{op} \ .$$

Integrate this inequality and recall $\lim_{\epsilon \to 0} \frac{g_{\boldsymbol{\xi}}(\epsilon)}{\epsilon} = 0$, we get

$$\frac{g_{\boldsymbol{\xi}}(\epsilon)}{\epsilon} \leq \epsilon \cdot \sup_x \|\nabla \mathbf{b}^{\mu,\eta}(x)\|^2_{op} \ ,$$

which implies

$$g_{\boldsymbol{\xi}}(\epsilon) \leq \sup_x \|\nabla \mathbf{b}^{\mu,\eta}(x)\|_{op}^2 \cdot \epsilon^2 \int \|\boldsymbol{\xi}(x)\|^2 \mathrm{d}\mu_t(x) = \sup_x \|\nabla \mathbf{b}^{\mu,\eta}(x)\|_{op}^2 \cdot W_2^2(\mu_t, \nu) \ .$$

Recall the definition of $g_{\boldsymbol{\xi}}(\epsilon)$, we have shown for any $\nu \in \mathcal{P}_2^r(X)$,

$$W_2^2(\mathbf{b}_{\#}^{\mu,\eta}\mu_t, \mathbf{b}_{\#}^{\mu,\eta}\nu) \leq g_{\boldsymbol{\xi}}(\epsilon) \leq \sup_x \|\nabla \mathbf{b}^{\mu,\eta}(x)\|_{op}^2 \cdot W_2^2(\mu_t, \nu) \ . \tag{18}$$

Therefore, we have for any $\nu \in \mathcal{P}_2^r(X)$

$$\frac{W_2^2(\mathbf{b}_{\#}^{\mu,\eta}\mu_t, \mathbf{b}_{\#}^{\mu,\eta}\nu) - W_2^2(\mu_t, \nu)}{W_2^2(\mu_t, \nu)} \leq \sup_x \|\nabla \mathbf{b}^{\mu,\eta}(x)\|_{op}^2 - 1 \ .$$

To further bound the right-hand side, for $\eta$ small enough, we notice

$$\nabla \mathbf{b}^{\mu,\eta}(x) = I_d + \eta(\nabla^2 f(x) + \beta^{-1}\nabla^2 \log p_{\mu_t}(x)) \preceq \left(1 + \eta(\kappa - \beta^{-1}\zeta)\right)I_d \ .$$

We complete the proof by taking the $\eta \to 0$ limit. $\qquad \square$

*Proof of Corollary 1.* First note

$$\nabla \mathbf{b}^{\mu,\eta}(x) = I_d + \eta(\nabla^2 f(x) + \beta^{-1}\nabla^2 \log p_{\mu_t}(x)) = \left(1 + \eta(\kappa - \beta^{-1}\zeta)\right)I_d \ ,$$
$$\mathbf{b}^{\mu,\eta} = (1 + \eta(\kappa - \beta^{-1}\zeta))\mathbf{i}, \quad \mathbf{f} := (\mathbf{b}^{\mu,\eta})^{-1} = (1 + \eta(\kappa - \beta^{-1}\zeta))^{-1}\mathbf{i} \ .$$

Define for convenience, $\mu_{t-\eta} := \mathbf{b}_{\#}^{\mu,\eta}\mu_t, \nu_{-\eta} := \mathbf{b}_{\#}^{\mu,\eta}\nu$, then in the proof of Theorem 2, (18) already proved

$$\frac{W_2^2(\mu_t, \nu)}{W_2^2(\mathbf{b}_{\#}^{\mu,\eta}\mu_t, \mathbf{b}_{\#}^{\mu,\eta}\nu)} = \frac{W_2^2(\mathbf{f}_{\#}\mu_{t-\eta}, \mathbf{f}_{\#}\nu_{-\eta})}{W_2^2(\mu_{t-\eta}, \nu_{-\eta})} \leq \sup_x \|\nabla \mathbf{f}(x)\|_{op}^2 = (1 + \eta(\kappa - \beta^{-1}\zeta))^{-2} \ .$$

Thus taking the reciprocal, then taking the limit inferior as $\eta \to 0$, we obtain

$$\liminf_{\eta \to 0} \frac{1}{\eta} \frac{W_2^2(\mathbf{b}_{\#}^{\mu,\eta}\mu_t, \mathbf{b}_{\#}^{\mu,\eta}\nu) - W_2^2(\mu_t, \nu)}{W_2^2(\mu_t, \nu)} \geq 2(\kappa - \beta^{-1}\zeta) \ .$$

Theorem 2 already proved

$$\limsup_{\eta \to 0} \frac{1}{\eta} \frac{W_2^2(\mathbf{b}_{\#}^{\mu,\eta}\mu_t, \mathbf{b}_{\#}^{\mu,\eta}\nu) - W_2^2(\mu_t, \nu)}{W_2^2(\mu_t, \nu)} \leq 2(\kappa - \beta^{-1}\zeta) \ .$$

Thus, the equality holds. $\qquad \square$

*Proof of Theorem 3.* Note by definition of the diffuse-then-denoise step, we have

$$W_2^2(\mathbf{b}_{\#}^{\mu,\eta}\mathbf{f}_{\#}^{\mu,\eta}\mu, \mu)$$
$$\leq \int \|\mathbf{b}^{\mu,\eta} \circ \mathbf{f}^{\mu,\eta}(x) - x\|^2 \mathrm{d}\mu \ ,$$
$$= \int \|\mathbf{f}^{\mu,\eta}(x) + \eta(\nabla f(\mathbf{f}^{\mu,\eta}(x)) + \beta^{-1}\nabla \log p_{\mu_\eta}(\mathbf{f}^{\mu,\eta}(x))) - x\|^2 \mathrm{d}\mu \ ,$$
$$= \int \|\mathbf{f}^{\mu,\eta}(x) + \eta \nabla f(\mathbf{f}^{\mu,\eta}(x)) + \eta\beta^{-1}\nabla \log p_\mu(x) + O(\eta^2) - x\|^2 \mathrm{d}\mu \ ,$$
$$= \int \|x - \eta\nabla f(x) - \eta\beta^{-1}\nabla \log p_\mu(x) + \eta\nabla f(x) + \eta\beta^{-1}\nabla \log p_\mu(x) + O(\eta^2) - x\|^2 \mathrm{d}\mu \ ,$$
$$= O(\eta^4) \ ,$$

where the second to the third line is applying the change of variable formula

$$\log p_{\mu_\eta}(\mathbf{f}^{\mu,\eta}(x)) = \log p_\mu(x) - \log \det\left(I_d - \eta(\nabla^2 f(x) + \beta^{-1}\nabla^2 \log p_\mu(x))\right),$$
$$= \log p_\mu(x) + \eta\, tr(\nabla^2 f(x) + \beta^{-1}\nabla^2 \log p_\mu(x)) + O(\eta^2),$$

and the third to the fourth line uses the fact

$$\nabla f(\mathbf{f}^{\mu,\eta}(x)) = \nabla f(x) + \eta\nabla^2 f(x)(\nabla f(x) + \beta^{-1}\nabla \log p_\mu(x)) + O(\eta^2).$$

The proof is completed. $\square$

*Proof of Corollary 2.* Without loss of generality, assume $W_2(\mu,\nu) = 1$. By Theorem 1, we know

$$W_2(\mu_{K\eta}, \nu_{K\eta}) \leq 1 - \eta K + O(\eta^2) \leq \exp\left(-\eta K + O(\eta^2)\right).$$

Using Theorem 2

$$\frac{W_2((\mathbf{b}_K^{\mu;\eta})_\#\mu_{K\eta}, (\mathbf{b}_K^{\mu;\eta})_\#\nu_{K\eta})}{W_2(\mu_{K\eta}, \nu_{K\eta})} \leq 1 + \eta(1 - \beta^{-1}\zeta_{K\eta}) + O(\eta^2),$$
$$\leq \exp\left(\eta(1 - \beta^{-1}\zeta_{K\eta}) + O(\eta^2)\right),$$
$$W_2((\mathbf{b}_K^{\mu;\eta})_\#\mu_{K\eta}, (\mathbf{b}_K^{\mu;\eta})_\#\nu_{K\eta})) \leq \exp\left(-\eta(K-1) - \eta\beta^{-1}\zeta_{K\eta} + O(\eta^2)\right).$$

Now recall Theorem 3, we have

$$W_2(\mu_{(K-1)\eta}, (\mathbf{b}_K^{\mu;\eta})_\#\mu_{K\eta}) = W_2(\mu_{(K-1)\eta}, (\mathbf{b}_K^{\mu;\eta} \circ \mathbf{f}_{K-1}^{\mu,\eta})_\#\mu_{(K-1)\eta}) = O(\eta^2),$$

and thus

$$W_2(\mu_{(K-1)\eta}, (\mathbf{b}_K^{\mu;\eta})_\#\nu_{K\eta})) \leq W_2(\mu_{(K-1)\eta}, (\mathbf{b}_K^{\mu;\eta})_\#\mu_{K\eta}) + W_2((\mathbf{b}_K^{\mu;\eta})_\#\mu_{K\eta}, (\mathbf{b}_K^{\mu;\eta})_\#\nu_{K\eta})),$$
$$\leq \exp\left(-\eta(K-1) - \eta\beta^{-1}\zeta_{K\eta} + O(\eta^2)\right).$$

Repeat the same argument, we know

$$W_2\left(\mu_{(K-2)\eta}, (\mathbf{b}_{K-1}^{\mu;\eta} \circ \mathbf{b}_K^{\mu;\eta})_\#\nu_{K\eta})\right) \leq \exp\left(-\eta(K-2) - \eta\beta^{-1}(\zeta_{(K-1)\eta} + \zeta_{K\eta}) + O(\eta^2)\right).$$

Chaining this bound recursively, we have

$$W_2\left(\mu, \left(\mathbf{b}_{[K]}^\mu \circ \mathbf{f}_{[K]}^\nu\right)_\#\nu\right) = W_2(\mu, (\mathbf{b}_{[K]}^\mu)_\#\nu_{K\eta}) \leq \exp\left(-\eta\beta^{-1}\sum_{k=1}^K \zeta_{k\eta} + O(\eta^2)\right).$$

$\square$

## A.3 Proofs for Section 4

*Proof of Proposition 5.* Recall the definition

$$\frac{\mathrm{d}m_\mu(\delta, \mathsf{r})}{\mathrm{d}\delta} = \frac{s_\mathsf{r}(1-\delta)\cdot\delta - \int_{1-\delta}^\infty s_\mathsf{r}(u)\mathrm{d}u}{\delta^2}.$$

(1) In the log-concave case, we know $s_\mathsf{r}(1) = 0$, therefore

$$\frac{\mathrm{d}m_\mu(\delta, \mathsf{r})}{\mathrm{d}\delta} = \frac{s_\mathsf{r}(1-\delta)\cdot\delta - \int_{1-\delta}^1 s_\mathsf{r}(u)\mathrm{d}u}{\delta^2} = \frac{\int_{1-\delta}^1 [s_\mathsf{r}(1-\delta) - s_\mathsf{r}(u)]\mathrm{d}u}{\delta^2} \geq 0$$

which shows the non-decreasing shape of $m_\mu(\cdot, \mathsf{r})$. (2) In the non-log-concave case, we have $s_\mathsf{r}(1) > 0$, and thus

$$\frac{\mathrm{d}m_\mu(\delta, \mathsf{r})}{\mathrm{d}\delta}\Big|_{\delta\to0+} < 0,$$

and we claim that (by taking the derivative w.r.t $\delta$)

$$\delta \mapsto s_r(1-\delta) \cdot \delta - \int_{1-\delta}^{\infty} s_r(u)\mathrm{d}u$$

is non-decreasing in $\delta$. Therefore, $\delta \mapsto \frac{\mathrm{d}m_\mu(\delta,r)}{\mathrm{d}\delta}$ either crosses zero (in a non-decreasing way) or stays negative for $\delta \in [0,1]$. Therefore, $m_\mu(\delta,r)$ is either (i) non-increasing in $\delta \in (0,1]$, or (ii) U-shaped in $\delta \in (0,1]$, namely first non-increasing then non-decreasing. $\square$

*Proof of Theorem 4.* As in the proof of Theorem 2, for any $\nu \in \mathcal{M}(\mu_t, M)$, define two quantities

$$\epsilon := \|\mathbf{t}_{\mu_t}^\nu - \mathbf{i}\|_{L^2(\mu_t;\mathbb{R}^d)} = W_2(\nu, \mu_t) \ ,$$
$$\boldsymbol{\xi} := (\mathbf{t}_{\mu_t}^\nu - \mathbf{i})/\|\mathbf{t}_{\mu_t}^\nu - \mathbf{i}\|_{L^2(\mu_t;\mathbb{R}^d)} \ ,$$

Then $\mathbf{t}_{\mu_t}^\nu = \mathbf{i} + \epsilon\boldsymbol{\xi}$, and as $\nu \xrightarrow{W_2} \mu_t$, we know $\epsilon \to 0$. Now note additionally if we assume $\nu \in \mathcal{M}(\mu_t, M)$, then the corresponding $\boldsymbol{\xi}$ satisfies

$$\boldsymbol{\xi} \in \mathcal{T}_{\mu_t}(M) := \left\{ \boldsymbol{\xi} \ : \ \int \|\boldsymbol{\xi}\|^2 \mathrm{d}\mu_t = 1, \ \sup_{x \in \mathrm{Dom}(\mu_t)} \|\boldsymbol{\xi}(x)\|^2 \leq M \right\} \ .$$

Claim that for any sequence of $\nu \in \mathcal{P}_2^r(X) : \nu \xrightarrow{W_2} \mu_t$ that attains the limit superior, we have

$$\limsup_{\nu \in \mathcal{M}(\mu_t,M):\nu \xrightarrow{W_2} \mu_t} \frac{W_2^2(\mathbf{b}_{\#}^{\mu,\eta}\mu_t, \mathbf{b}_{\#}^{\mu,\eta}\nu) - W_2^2(\mu_t, \nu)}{W_2^2(\mu_t, \nu)}$$
$$\leq \sup_{\boldsymbol{\xi} \in \mathcal{T}_{\mu_t}(M)} \frac{\int \|\nabla\mathbf{b}^{\mu,\eta}(x)\boldsymbol{\xi}(x)\|^2 \mathrm{d}\mu_t(x) - \int \|\boldsymbol{\xi}(x)\|^2 \mathrm{d}\mu_t(x)}{\int \|\boldsymbol{\xi}(x)\|^2 \mathrm{d}\mu_t(x)} \ .$$

To derive this claim, notice that

$$W_2^2(\mathbf{b}_{\#}^{\mu,\eta}\mu_t, \mathbf{b}_{\#}^{\mu,\eta}\nu) \leq \int \|\mathbf{b}^{\mu,\eta} \circ (\mathbf{i} + \epsilon\boldsymbol{\xi}) - \mathbf{b}^{\mu,\eta} \circ \mathbf{i}\|^2 \mathrm{d}\mu_t \ ,$$
$$= \int \|\mathbf{b}^{\mu,\eta}(x + \epsilon\boldsymbol{\xi}(x)) - \mathbf{b}^{\mu,\eta}(x)\|^2 \mathrm{d}\mu_t(x) \ ,$$
$$= \epsilon^2 \left( \int \|\nabla\mathbf{b}^{\mu,\eta}(x)\boldsymbol{\xi}(x)\|^2 \mathrm{d}\mu_t(x) + o_\epsilon(1) \right) \ .$$

The last step requires some justification. Define an auxiliary function $g(\epsilon) := \int \|\mathbf{b}^{\mu,\eta}(x + \epsilon\boldsymbol{\xi}(x)) - \mathbf{b}^{\mu,\eta}(x)\|^2 \mathrm{d}\mu_t(x)$, we can verify that

$$g'(0) = 0,$$
$$g''(0) = 2\int \|\nabla\mathbf{b}^{\mu,\eta}(x)\boldsymbol{\xi}(x)\|^2 \mathrm{d}\mu_t(x) \ ,$$
$$g''(\epsilon) = 2\int \|\nabla\mathbf{b}^{\mu,\eta}(x + \epsilon\boldsymbol{\xi}(x))\boldsymbol{\xi}(x)\|^2 \mathrm{d}\mu_t(x)$$
$$+ 2\int \left\langle \nabla^2\mathbf{b}^{\mu,\eta}(x + \epsilon\boldsymbol{\xi}(x)), \boldsymbol{\xi}(x) \otimes \boldsymbol{\xi}(x) \otimes \left(\mathbf{b}^{\mu,\eta}(x + \epsilon\boldsymbol{\xi}(x)) - \mathbf{b}^{\mu,\eta}(x)\right) \right\rangle \ .$$

By the Taylor's Theorem, there exists $\tilde{\epsilon} \in [0, \epsilon]$, such that

$$g(\epsilon) = \frac{1}{2}g''(\tilde{\epsilon})\epsilon^2 \ .$$

Notice $g''(\tilde{\epsilon}) = g''(0) + o_\epsilon(1)$ due to the fact that $\|\boldsymbol{\xi}(x)\|^2 \leq M$ uniformly bounded and that $\mathbf{b}^{\mu,\eta}$ has bounded derivatives, we establish the claim.

Now we proceed to control the term $\int \|\nabla \mathbf{b}^{\mu,\eta}(x)\boldsymbol{\xi}(x)\|^2 \mathrm{d}\mu_t(x)$

$$\int \|\nabla \mathbf{b}^{\mu,\eta}(x)\boldsymbol{\xi}(x)\|^2 \mathrm{d}\mu_t(x) \leq \int \|\nabla \mathbf{b}^{\mu,\eta}(x)\|_{op}^2 \|\boldsymbol{\xi}(x)\|^2 \mathrm{d}\mu_t(x) \ .$$

Now let's study $\|\nabla \mathbf{b}^{\mu,\eta}(x)\|_{op}$: recall $\mathsf{s}(t) = \sqrt{\beta^{-1}(1 - e^{-2t})}$

$$\begin{aligned}
\|\nabla \mathbf{b}^{\mu,\eta}(x)\|_{op} &= \|I_d + \eta I_d - \tfrac{\eta}{1-e^{-2t}}(I_d - Cov[\mathsf{r}(t)\mathbf{X}_0|\mathbf{X}_t = x])\|_{op} \\
&= (1+\eta) - \tfrac{\eta}{1-e^{-2t}} + \tfrac{\eta}{1-e^{-2t}}\| \, Cov[\mathsf{r}(t)\mathbf{X}_0|\mathbf{X}_t = x]\|_{op} \\
&= (1+\eta) - \tfrac{\eta}{1-e^{-2t}} + \tfrac{\eta}{1-e^{-2t}} L_{\mathsf{r}(t)}(\tfrac{x}{\mathsf{s}(t)})
\end{aligned}$$

Here we recall the conditional covariance (localization function) defined before

$$L_{\mathsf{r}}(y) := \| \, Cov[\mathsf{r}\mathbf{X}|\mathbf{Y}_{\mathsf{r}} = y]\|_{op} \ .$$

Continue, we have

$$\int \|\nabla \mathbf{b}^{\mu,\eta}(x)\boldsymbol{\xi}(x)\|^2 \mathrm{d}\mu_t(x) - \int \|\boldsymbol{\xi}(x)\|^2 \mathrm{d}\mu_t(x)$$

$$\leq 2\eta \int \|\boldsymbol{\xi}(x)\|^2 \mathrm{d}\mu_t(x) - \tfrac{2\eta}{1-e^{-2t}} \int \|\boldsymbol{\xi}(x)\|^2 \mathrm{d}\mu_t(x)$$

$$+ \tfrac{2\eta}{1-e^{-2t}} \int L_{\mathsf{r}(t)}(\tfrac{x}{\mathsf{s}(t)})\|\boldsymbol{\xi}(x)\|^2 \mathrm{d}\mu_t(x) + O(\eta^2) \ .$$

We analyze the last term: first, we define the region

$$R_\delta := \{x \in X \ : \ L_{\mathsf{r}(t)}(\tfrac{x}{\mathsf{s}(t)}) \leq 1 - \delta\}$$

and bound the integral depending on the region,

$$\begin{aligned}
\int L_{\mathsf{r}(t)}(\tfrac{x}{\mathsf{s}(t)})\|\boldsymbol{\xi}(x)\|^2 \mathrm{d}\mu_t(x) &= \int_{x \in R_\delta} L_{\mathsf{r}(t)}(\tfrac{x}{\mathsf{s}(t)})\|\boldsymbol{\xi}(x)\|^2 \mathrm{d}\mu_t(x) \\
&\quad + \int_{x \in R_\delta^c} L_{\mathsf{r}(t)}(\tfrac{x}{\mathsf{s}(t)})\|\boldsymbol{\xi}(x)\|^2 \mathrm{d}\mu_t(x) \ , \\
&\leq (1-\delta)\left( \int_{x \in R_\delta} \|\boldsymbol{\xi}(x)\|^2 \mathrm{d}\mu_t(x) + \int_{x \in R_\delta^c} \|\boldsymbol{\xi}(x)\|^2 \mathrm{d}\mu_t(x) \right) \\
&\quad + \int_{x \in R_\delta^c} \left(L_{\mathsf{r}(t)}(\tfrac{x}{\mathsf{s}(t)}) - (1-\delta)\right)\|\boldsymbol{\xi}(x)\|^2 \mathrm{d}\mu_t(x) \ ,
\end{aligned}$$

Recalling $\boldsymbol{\xi} \in \mathcal{T}_{\mu_t}(M)$,

$$\begin{aligned}
&(1-\delta)\int \|\boldsymbol{\xi}(x)\|^2 \mathrm{d}\mu_t(x) + \int_{x \in R_\delta^c} \left(L_{\mathsf{r}(t)}(\tfrac{x}{\mathsf{s}(t)}) - (1-\delta)\right)\|\boldsymbol{\xi}(x)\|^2 \mathrm{d}\mu_t(x) \\
&= (1-\delta)\int \|\boldsymbol{\xi}(x)\|^2 \mathrm{d}\mu_t(x) + \int \|\boldsymbol{\xi}(x)\|^2 \mathrm{d}\mu_t(x) \cdot M \cdot \int_{x \in R_\delta^c} \left(L_{\mathsf{r}(t)}(\tfrac{x}{\mathsf{s}(t)}) - (1-\delta)\right)\mathrm{d}\mu_t(x) \ , \\
&= \int \|\boldsymbol{\xi}(x)\|^2 \mathrm{d}\mu_t(x) \cdot \left((1-\delta) + M \cdot \int_{1-\delta}^{\infty} s_{\mathsf{r}(t)}(z)\mathrm{d}z\right) \ ,
\end{aligned}$$

where the last step uses Proposition 4 and the Fubini's theorem: for a non-negative random variable $Z > 0$, if $\mathbb{E}[Z] < \infty$, then $\int_C^\infty (z - C)p_Z(z)\mathrm{d}z = \int_C^\infty P(Z \geq z)\mathrm{d}z$ for $C > 0$; set $Z = L_{\mathsf{r}(t)}(\tfrac{\mathbf{X}_t}{\mathsf{s}(t)})$ and $C = 1 - \delta$. Put things together, for any $\delta \in [0,1]$

$$\sup_{\boldsymbol{\xi} \in \mathcal{T}_{\mu_t}(M)} \frac{\int \|\nabla \mathbf{b}^{\mu,\eta}(x)\boldsymbol{\xi}(x)\|^2 \mathrm{d}\mu_t(x) - \int \|\boldsymbol{\xi}(x)\|^2 \mathrm{d}\mu_t(x)}{\int \|\boldsymbol{\xi}(x)\|^2 \mathrm{d}\mu_t(x)}$$

$$\leq 2\eta - \tfrac{2\eta}{1-e^{-2t}}\left[\delta - M \cdot h_\mu(\delta, \mathsf{r}(t))\right] + O(\eta^2) \ .$$

$\square$

