# OpenReview forum: "Denoising Diffusions with Optimal Transport: Localization, Curvature, and Multi-Scale Complexity"
_TMLR — Accepted by TMLR_

### Review · Reviewer_sMiA · 2025-11-11

**Summary Of Contributions:**

The paper studies the denoising process, that is, the reverse process in the diffusion model. In particular, the paper studies the Wasserstein contraction that is characterized by the curvature complexity of a smoothed data distribution as a specific signal-to-noise ratio scale. The multi-scale curvature complexity determines the difficulty of the denoising chain. The multi-scale complexity quantifies a notion of average-case curvature instead of the worst-case scenario. It depends on the integrated tail function, measuring the relative mass of locations with positive curvature versus those with negative curvature. Several examples are given.

**Audience:**

Yes

**Audience Explanation:**

I believe so. I think there has been a lot of interest in generative models such as a diffusion model in recent years. Various convergence guarantees have been obtained in the literature, mostly in TV and KL divergence, and some papers deal with Wasserstein convergence. But the papers that I am aware of that deal with Wasserstein convergence usually need to assume some very strong assumptions, such as strongly-log-concave data distributions. This paper introduces some new ideas to use a notion of average-case curvature instead of the worst-case scenario to measure the contraction of the reverse process. This might lead to some future research in this field.

**Broader Impact Concerns:**

N/A.

**Claims And Evidence:**

Yes

**Claims Explanation:**

This is a theoretical paper. The proof seems to be rigorous. The paper is relatively well-written.

**Requested Changes:**

(1) On page 2, you wrote that most of the current theoretical literature (Lee et al., 2023; Chen et al., 2022;2023) treat this curvature as a nuisance by, for example, leveraging a form of data-processing inequality and only focusing on non-expansion metrics $d$... for $d\in\{d_{TV},d_{KL}\}$. In your paper, you did not cite any theoretical works in Wasserstein convergence. You should cite and discuss the following papers.

Gao et. al. (2025). Wasserstein Convergence Guarantees for a General Class of Score-Based Generative Models. Journal of Machine Learning Research.

Bruno et al. (2025). On diffusion-based generative models and their error bounds: The log-concave case with full convergence estimates. Transactions on Machine Learning Research.

Both papers dealt with log-concave data distributions, and obtained contractions in Wasserstein distance.
There is another recent paper,

Silveri and Ocello (2025) Beyond Log-Concavity and Score Regularity: Improved Convergence Bounds for Score-Based Generative Models in W2-distance (ICML 2025),

that goes beyond the log-concave case. The field is developing very fast, and there might be other papers that derived contractions in Wasserstein distance. As a result, your sentence on page 7 that the current theory falls short in addressing the behavior of the backward denoising process under the Wasserstein metric is not completely accurate.

(2) After equation (5), you should introduce $\beta$ and explain the role it plays.

(3) The assumption in your Corollary 1 seems super strong. When you assume $\nabla^{2}f(x)=\kappa\cdot I_{d}$ and $\nabla^{2}\log p_{\mu_{t}}(x)=-\zeta\cdot I_{d}$, did you assume this holds for every $x$? If so, you are assuming the data distribution is Gaussian right?

(4) In the three papers I mentioned previously, sometimes, strong-log-concavity is only assumed for the data distribution, and then
the strong-log-concavity at any time $t$ can be derived as a result. In your results, it seems that you directly assume strong-log-concavity
at any time. At least, you should provide some discussions on that. For example, in your Corollary 2, you assumed strong-log-concavity coefficient $\zeta_{k\eta}$, but it is not clear how $\zeta_{k\eta}$ depends on $k\eta$.

(5) Your main results, such as Theorem 2 and Theorem 4, seem to be asymptotic in nature, i.e. requiring $\eta\rightarrow 0$.
It would be nice if you can extend that to non-asymptotic results, which are more relevant in deriving iteration complexity.
The current results are presented in an asymptotic way, which is one of the weaknesses of the paper.

(6) In Definition 6, please make $\{$ and $\}$ bigger.

(7) In Section 5.2, Beyond Log-Concavity, you presented a few special examples. I am wondering whether it is possible
to derive some general results beyond log-concavity. For example, instead of particular examples, how about
a general distribution that satisfies dissipativity condition or strong-log-concavity outside a compact domain?
That will significantly strengthen the contributions of the paper.

(8) Please move $+$ at the end of the first line in equation (13) to the beginning of the second line.

---

> ### Author Response · Authors · 2025-11-19
>
> Thank you for the comments. We address each point below.
>
> 1. Thanks for sharing these papers. We have discussed them in the revision - see Page 7, Section 1.2.
>
>     We would like to note that the work by Silveri and Ocello (2025) is concurrent work - ours is on arxiv in Nov 2024. Although the arguments are different, the Wasserstein analysis the authors undertake also lead to a very interesting characterization of contractive and non-contractive regimes during the backward denoising process. Their analysis is based on weak log-concavity assumptions similar to dissipativity: the target density is approximately log-concave outside a finite region/radius. Our work makes no such assumptions. Our characterization of contractive and non-contractive regimes is based on a notion of average curvature.
>
> 2. Here $\beta^{-1}$ serves as temperature. If $f(x)=x^2/2$ then the stationary distribution is $\rho_\infty \propto \exp(-\beta x^2/2)$. We have clarified this in the revision.
>
> 3. Yes. This construction is a lower bound (for sharpness of Theorem 1) demonstrating that the backward denoising map can be an expansion even if the density at time $t$ is log-concave (even Gaussian).
>
> 4. If the target distribution is log-concave, then log concavity holds at time $t$. Suppose $X$ and $Y$ are $\mu$/$\nu$ log-concave for $\mu, \nu\in \mathbb{R}_+$. Then $(X+Y)$ is $(\mu^{-1}+\nu^{-1})^{-1}$ log-concave (see Theorem 2.3 of Wellner 2013). This fact tracks the dependence of the log-concave parameter on $t$.
>
>     That said, we do not assume log-concavity at all times, $t$, in our results. For our backward denoising results (Theorem 2, Corollary 2), we only require worst-case curvature at time $t$ (or time $k\eta$) being bounded, as opposed to being negative. For example, in Corollary 2, we assume there exists some $\zeta_{k\eta} \in \mathbb{R}$ (positive or negative) such that $\nabla^2 \log p_{\mu_{k\eta}}(x) \preceq - \zeta_{k\eta} \cdot I_d, \ \forall x \in \mathcal{X},\  \forall k = 1,\ldots,K$. Theorem 4 replaces this worst-case curvature $\zeta_{k\eta}$ by a notion of average curvature.
>
>     The only place we make a restriction on log-concavity, $\zeta > 0$, is in Corollary 1. However, this is a lower bound result intended to show the sharpness of Theorem 2. See point (3).
>
> 5. We can extend the results in Theorems 2 and 4 for non-asymptotic $\eta$:
>
>     By the assumption in Theorem 2, the eigenvalues of $\nabla^2 f(x)$ and $\nabla^2 \log p_{\mu_t}(x)$ are uniformly bounded. Then, we can take a positive constant $K>0$ such that
>     $-K \preceq  \nabla^2 f(x) + \beta^{-1} \nabla^2 \log p_{\mu_t}(x)$, and hence the Jacobian of the backward map $b^{\mu, \eta}$ can be uniformly controlled as $1 - \eta K \preceq \nabla b^{\mu, \eta}(x)\preceq 1+ \eta (\kappa-\beta^{-1}\zeta)$.
>     Thus, for all $\eta<1/K$, $b^{\mu, \eta}(x)$ is $1+\eta (\kappa-\beta^{-1}\zeta)$-Lipschitz so that
>
>     $$
>     \frac{W_2( b^{\mu, \nu} \mu_t,  b^{\mu, \nu} \nu )}{W_2(\mu_t, \nu)}   \le  1 + \eta (\kappa - \beta^{-1} \zeta) \quad \text{for all $\eta<1/K$} \quad \text{(nonasymptotic version of Theorem 2)}
>     $$
>
>     For Theorem 4, we can modify the proof similarly to make it work for non-asymptotic $\eta$. We choose to present the asymptotic version for simplicity of presentation.
> 6. Fixed.
> 7. It is possible to extend our analysis; we say a function $f:\mathbb{R}^d \rightarrow \mathbb{R}^d$ is $M$-smooth and $(m, b)$-dissipative if
>     $$
>     \|\nabla f(y) - \nabla f(x)\| \leq M \|y - x\| \quad \text{and} \quad
>     \langle \nabla f(y) - \nabla f(x), y - x \rangle \geq m \|y - x\|^2 - b \quad \text{for all $x, y\in \mathbb{R}^d$}
>     $$
>
>     Assume that $y \mapsto -\log p(y)$ is $(m, b)$-dissipative and $M$-smooth, then letting $b(y)=(1+\eta) y + \beta^{-1} \eta \nabla \log p_{\mu_t}(y)$ be the backward map, we have
>     $$
>     W^2(b_\sharp \mu_t, b_\sharp \nu)  \leq 2(1+\eta)\eta \beta^{-1} b + \left[ (1+\eta - \eta \beta^{-1} m)^2 + \eta \beta^{-1} (M^2 - m^2) \right] W^2(\mu_t, \nu)
>     $$
>     Proof: By the definition of Wasserstein distance, letting $\mathbb{E}$ be the expectation under the optimal coupling of $(X, Y)$,
>     $$
>     W^2(b_\sharp \mu_t, b_\sharp \nu)  \leq \mathbb{E} \|b(X) - b(Y)\|^2
>     = \mathbb{E} \| (1+\eta)(X - Y) + \beta^{-1} \eta (\nabla \log p(X) - \nabla \log p(Y)) \|^2
>     $$
>     Expanding the square, the RHS reads
>     $$
>     (1+\eta)^2 W_2(\mu_t, \nu)^2 + (\beta^{-1} \eta)^2 \mathbb{E} \|\nabla \log p(X) - \nabla \log p(Y)\|^2 - 2 (1+\eta) \beta^{-1} \eta \mathbb{E} \langle X - Y, \nabla -\log p(X) + \nabla \log p(Y) \rangle.
>     $$
>     Bounding the second term by the M-smoothness and the third term by the dissipativity, we obtain the claim.
>
>     Note this dissipative approach ignores the fine-grained structure of the density $p_t$, and only relies on global smoothness and dissipativity $\nabla^2 \log p_t$. Our average curvature combines density $p_t$ and curvature $\nabla^2 \log p_t$ together.
> 8. Fixed.

---

> > ### Comment · Reviewer_sMiA · 2025-12-09
> > **response**
> >
> > I thank the author(s) who provided a detailed point-to-point response to the questions and requests I raised in the previous round. I do not have further comments at this stage.

---

### Review · Reviewer_r25w · 2025-11-13

**Summary Of Contributions:**

This paper provides a rigorous theoretical framework for analyzing the difficulty of denoising diffusion models by connecting them to Wasserstein-2 (W2) Optimal Transport (OT) and information geometry. The paper's main claim is that the difficulty of the reverse process is governed by the local curvature of the smoothed data distribution $p_t$ at different signal-to-noise ratio (SNR) scales.

The paper’s core contributions are the following:

**1. Score Function as W2-Gradient Flow:**
The paper first proves that the denoising score function, $\nabla \log p_t(x)$, is the velocity field of the reverse diffusion process, which can be framed as a W2-gradient flow of the KL divergence (known as the Benamou-Brenier formulation).

**2. Curvature (Hessian) Governs Localization Uncertainty:**
The "difficulty" of a single denoising step—defined as the localization uncertainty (i.e., the conditional variance $\text{Var}(x_{t-1} | x_t)$)—is proven to be governed by the local curvature, specifically the Hessian of the log-density, $\nabla^2 \log p_t(x)$.

*Negative Curvature (Log-Concave Regions):* $\nabla^2 \log p_t \prec 0$. These regions are stable, "contractive," and easy to denoise (low conditional variance).

*Positive Curvature (Non-Log-Concave Regions):* $\nabla^2 \log p_t \succ 0$. These regions are "expansive," unstable, and hard to denoise (high conditional variance).

**3. Multi-Scale Curvature Complexity:**
The paper introduces a new metric that quantifies the total generative difficulty. This metric is an average-case measure that integrates the positive (expansive) part of the curvature across all SNR scales. This metric is crucial for identifying the "bottleneck SNR"—the specific noise scale $t$ where the average positive curvature is highest, representing the most difficult part of the generative process.

Key Illustrations

The authors provide extensive illustrations of their result using Gaussian Mixture Models (GMMs):

*Curvature Analysis:*

The paper shows that the geometry of the denoising task changes dramatically with the noise scale $t$:

Specifically they show that within modes, the high-probability regions within each individual Gaussian mode are log-concave (negative curvature). These regions are "easy" to denoise. However, the low-probability "ridge" or "trench" between the modes is non-log-concave (high positive curvature). This region is "hard" to denoise, as the model must learn to "split" the noise and "pull" data points back towards their respective modes, creating an "expansive" flow.

They then show that the "Multi-Scale Curvature Complexity" metric successfully identifies the most difficult part of the generative process, and this sits between the low noise as well as high noise regimes, specifically the "bottleneck" occurs at the specific intermediate noise level where the noise is just enough to "lift" the low-probability ridge, causing the modes to merge. At this scale, the unstable positive-curvature region is at its most prominent and widespread before being completely washed out by noise.

**Audience:**

Yes

**Audience Explanation:**

This paper focuses on theoretically understanding the complexity of the denoising process of a diffusion model. The paper sheds light on the difficulty of denoising at various SNR scales, and I feel this is of particular interest to members of the machine learning community at this point.

**Broader Impact Concerns:**

No broader impact concerns.

**Claims And Evidence:**

Yes

**Claims Explanation:**

The paper’s claims are accurate and they provide detailed proofs. I have skimmed these and the proofs seem correct to me.

**Requested Changes:**

I feel the paper would be significantly strengthened by improving clarity of the presentation. It would be nice to have, for instance, a clear section defining notation and perhaps in the appendix a section at least stating several properties and theorems that have been just stated in the body.

---

> ### Author Response · Authors · 2025-11-19
>
> Thank you for the comment, and we revised the paper to clarify. We have a section for notation - paragraph 1 in Section 1.1, which we have now emphasized in bold text. As suggested, we included a paragraph in the appendix stating properties and theorems that are established. See below
>
> "The Appendix is organized as follows.
>
> Section A.1 gathers the technical preparations for denoising and localization, where we prove the results in Section 2. We derive the optimal transport denoising map and explain how the curvature function appears from a localization perspective.
>
> Section A.2 presents analyses of the forward and backward processes, where we prove the results in Section 3. We demonstrate how the curvature function influences backward denoising via a sensitivity analysis under the Wasserstein metric. We also analyze the diffuse-then-denoise process, showing that adding noise then denoising can be beneficial in the presence of negative curvature.
>
> Section A.3 provides proofs for the results in Section 4. We extend beyond the log-concave setting by defining and analyzing a multi-scale complexity that captures an average notion of curvature across all time scales. This multi-scale complexity plays a key role in the sensitivity analysis of the diffusion model under the Wasserstein metric."

---

> > ### Comment · Reviewer_r25w · 2025-12-19
> >
> > Thank you for your response. I have no other questions at this stage.

---

### Review · Reviewer_AQYj · 2025-11-14

**Summary Of Contributions:**

This paper presents a sensitivity analysis of diffusion-based generative models, examining their stability under the Wasserstein-2 ($W_2$) metric. Consider a standard forward diffusion process that transforms a data distribution $\mu$ into a noised distribution $\mu_{K\eta}$ at time $K\eta$, the author investigates the behavior of a backward denoising process initiated from a perturbed distribution $\nu \approx \mu_{K\eta}$. This analysis tracks the evolution of the $W_2$ distance between the distributions of this perturbed chain ($\nu_{k\eta}$) and the true backward chain ($\mu_{k\eta}$) at each step $k$.

The primary contribution is demonstrating that the final error between the generated distribution $\nu_0$ and the target data distribution $\mu$ (i.e., $W_2(\mu, \nu_0)$) is fundamentally governed by the "multi-scale complexity," a quantity defined by the average-case curvature of the log-density of the intermediate distributions, $\nabla^2 \log p_{k\eta}(x)$, under the distribution $p_{k\eta}$. The authors also compare this complexity across different signal-to-noise ratios, illustrating their findings with several specific examples.

Strength:

1. The paper provides a novel stability analysis of the backward denoising process using the Wasserstein-2 metric. This approach extends the existing theoretical framework for diffusion models, capturing more detailed behavior than previous analyses based on KL divergence or TV distance. A key benefit is that this $W_2$ analysis goes beyond worst-case scenarios and provides tight characterizations of the model's stability.

2. The findings of this paper could have practical value. The connection between "multi-scale complexity" (i.e., average-case curvature) and stability at different time steps may offer a useful guide for designing and scheduling the noise parameters in diffusion models.

Weakness:

1. The paper's conceptual descriptions are at times overly stylized, which can hinder a clear understanding.

2. The focus on the "average-case behavior" of curvature is not an entirely novel concept. This idea has been explored in prior literature on diffusion model analysis, and its connection to stability has been previously discussed, even though they stems from a different setup (discretization analysis).

**Audience:**

Yes

**Audience Explanation:**

The theoretical analysis of diffusion models is a highly active and popular topic within the machine learning theory community, which forms a key part of TMLR's audience. This paper contributes several interesting observations and a novel analytical framework to this specific area. The findings will likely be of significant interest to researchers working on the foundations of generative models.

**Broader Impact Concerns:**

None.

**Claims And Evidence:**

Yes

**Claims Explanation:**

The main theoretical claim is justified by mathematically rigorous analysis.

**Requested Changes:**

1. Clarify conceptual sentences in the Introduction. Some of the framing is ambiguous and not convincing. Here are two examples

(1)The paper states: "Consequently, $d(\mu_0, \bar{\nu}_0) \le d(\mu_K, \nu)$ is solely determined by the contraction of the forward diffusion chain alone." This is confusing, as the bound $d(\mu_0, \bar{\nu}_0) \le d(\mu_K, \nu)$ in KL/TV analysis typically arises from applying the data-processing inequality to the backward chain (i.e., the denoising steps). It is unclear what is meant by this being "solely determined by the contraction of the forward diffusion chain."

(2)The paper states: “Therefore, justifying the diffusion model as a time reversal Markov chain aiming to recover the past from the future is an effort in vain. In contrast, we propose to study the diffusion models as a sensitivity analysis.” This is also a potentially confusing claim, as current theoretical studies often analyze the denoising process starting from a large time T (where the forward process is near-Gaussian), and the 'time reversal' perspective remains important in those studies. And those studies also provided some stability analysis of the backward process, though it's under a different metric.

The authors should revise these and similar sentences for clarity to avoid misunderstanding the setup and the paper's positioning. These  do not affect the overall evaluation, but it is important to make the paper clearer to the readers.

2.Add discussion on the related work regarding average-case curvature. The paper should give credit to previous papers that discuss the "average case curvature,"  (though they mainly aim to give upper bound)  such as https://arxiv.org/abs/2211.01916 and https://arxiv.org/abs/2308.03686. Moreover, the later work has improved the bound to $E \|\nabla^2 \log p_t(x)\|_F^2 \le d$, whereas this paper appears to use a naive bound.

---

> ### Author Response · Authors · 2025-11-19
>
> Thank you for the comments. We address each point below.
>
> 1(1). Recall $\bar{\nu}_K := \nu$, we mean that $d(\mu_0, \bar{\nu}_0)\le d(\mu_K, \bar{\nu}_K) = d(\mu_K, \nu)$ by the data-processing inequality, namely the backward chain is non-expansive. We know $d(\mu_K, \nu)\le c^K$ for a constant $c \in (0,1)$ which is determined by the contraction of forward to equilibrium $\nu$. Therefore, $d(\mu_0, \bar{\nu}_0)$ is bounded by the forward contraction term $c$ solely.
>
> 1(2). We have revised some sentences for clarity. For example, we have revised the sentence as below: "Therefore, framing the diffusion model as a time-reversal Markov chain—one that recovers the past from the future—does not fully resolve the underlying conceptual issues."
>
> We meant to say that sensitivity analysis is crucial, as one cannot recover the initial distribution from a Gaussian distribution at $T = \infty$, following the time reversal chain. Some previous papers are in agreement with this viewpoint and thus study perturbations to the time reversal chain at a finite $T$, and derive bounds under metrics satisfying data-processing inequalities. However, due to the nature of data-processing inequalities, the fine-grained property of the backward denoising maps is not directly revealed, even if the score function is known exactly; namely, those metrics are insensitive to the choice of backward denoising maps.
>
> We study whether this perturbation is exacerbated during the backward denoising process, under the Wasserstein metric. This question is central to our analysis and motivated us to investigate whether the diffuse-then-denoise process (as a whole) is beneficial. We acknowledge that previous work noted that the backward denoising map may not be contractive, even for log-concave distributions. We pursue this line of thought further to show how average curvature plays the role.
>
> 2. We thank the reviewer for pointing out this literature, and we agree that it is important to discuss them in our paper.  Average notion of curvature with respect to the Frobenius norm (see, for instance, Benton et al., Conforti et al.) has been used to analyze the reverse SDE under Kullback-Leibler divergence. Our paper complements the literature by showing that a notion of average curvature also plays a role in convergence analysis when the metric is the Wasserstein distance. In contrast to the work you mention, our analysis of curvature is with respect to the operator norm, and is based on truncation arguments that highlight that the high-probability, negative-curvature regions of a density vs. the low-probability, positive-curvature regions determine convergence quality when the metric is the Wasserstein distance. This alternative analysis is central to why worst-case non-log-concavity is not necessarily a difficulty in diffusion models. We think our perspective helps bridge the parallel theoretical analyses of diffusion models in $f$-divergence and Wasserstein distance. Notions of average curvature are crucial to the quality of denoising diffusions, for metrics including KL and Wasserstein, and for processes including reverse SDEs and probability flow ODEs. Last, we only use the expectation bound---$E[tr( \nabla^2 \log p_t)] < 0$ and thus that $E[ Cov[\frac{X}{\sigma} | Y] \|_{op}]$ is finite from second-order Tweedie's formula---so that Fubini's theorem can be applied in a step of the analysis; this is clarified in the revised version.
>
>     We include these papers in our literature review; see the new paragraph on Page 7, Section 1.2.
>
>     "Notions of average curvature have been used to derive improved convergence bounds for diffusion models with respect to $d_{\rm KL}$ \citep{chen2023improved, benton2023nearly}. Our paper shows that a different notion of average curvature also plays a role in the sensitivity analysis when the metric is the Wasserstein distance. Our perspective helps bridge the parallel theoretical analyses of diffusion models under $f$-divergence and Wasserstein distance."

---

### Decision · Action_Editor_UXG2 · 2025-12-20

**Recommendation:** Accept as is

**Audience:**

Yes

**Audience Explanation:**

See above.

**Claims And Evidence:**

Yes

**Claims Explanation:**

I recommend acceptance of this manuscript. This paper provides a rigorous theoretical analysis of diffusion-based generative models through the lens of Wasserstein-2 optimal transport, introducing the novel concept of "multi-scale curvature complexity" to characterize denoising difficulty. The authors demonstrate that the effectiveness of the diffuse-then-denoise process is governed by average-case curvature rather than worst-case curvature, measured through an integrated tail function that balances regions of positive versus negative curvature across different signal-to-noise ratio scales. All three reviewers converged on acceptance after the authors thoroughly addressed raised concerns: they clarified the relationship to existing average-curvature analyses under KL divergence, added comprehensive discussion of concurrent and related work (Bruno et al., Gao et al., Gentiloni-Silveri & Ocello, Chen et al., Benton et al.), improved conceptual clarity in the introduction, provided non-asymptotic extensions of main results, and enhanced notation and presentation throughout. The work makes a significant theoretical contribution by bridging parallel analyses of diffusion models under f-divergence and Wasserstein distance, offering new insights into why non-log-concavity need not be problematic for denoising diffusions. The rigorous mathematical framework and illustrative examples using Gaussian mixture models will be valuable to TMLR's audience working on the foundations of generative models.